# Enhanced spatial clustering of single-molecule localizations with graph neural networks

Jesús Pineda [1], Sergi Masó-Orriols [2,3], Montse Masoliver[2,3], Joan Bertran[2,3], Mattias Goksör[1], Giovanni Volpe [1,4] ✉ & Carlo Manzo [2,3] ✉

Single-molecule localization microscopy generates point clouds corresponding to fluorophore localizations. Spatial cluster identification and analysis of these point clouds are crucial for extracting insights about molecular organization. However, this task becomes challenging in the presence of localization noise, high point density, or complex biological structures. Here, we introduce MIRO (Multifunctional Integration through Relational Optimization), an algorithm that uses recurrent graph neural networks to transform the point clouds in order to improve clustering efficiency when applying conventional clustering techniques. We show that MIRO supports simultaneous processing of clusters of different shapes and at multiple scales, demonstrating improved performance across varied datasets. Our comprehensive evaluation demonstrates MIRO's transformative potential for single-molecule localization applications, showcasing its capability to revolutionize cluster analysis and provide accurate, reliable details of molecular architecture. In addition, MIRO's robust clustering capabilities hold promise for applications in various fields such as neuroscience, for the analysis of neural connectivity patterns, and environmental science, for studying spatial distributions of ecological data.

The identification and analysis of clusters, i.e., data points sharing some similarity, are crucial across many scientific disciplines and technological applications. Clustering algorithms facilitate pattern recognition, data compression, and information retrieval, enabling researchers to uncover hidden structures within complex datasets. A notable application of clustering algorithms is the spatial analysis of single-molecule localization microscopy (SMLM) data[1–3]. Super-resolution techniques, such as stochastic optical reconstruction microscopy (STORM)[4], photoactivated localization microscopy (PALM)[5], points accumulation for imaging in nanoscale topography (PAINT)[6], and their variants, generate spatial point clouds, where each point represents the localization (typically with precision $\lesssim 20$ nm) of an individual molecule[7]. These datasets can contain millions of

localizations, which allows the application of statistical methods to provide detailed insights into the spatial organization of molecules within biological samples (Fig. 1a). Clustering SMLM data is crucial because it helps identify and group molecules that form specific cellular structures, such as protein nanoclusters[8–10], chromatin clutches[11], focal adhesions[12], or nuclear pore complexes[13]. By clustering these points, researchers can infer molecules' functional organization and interaction patterns under different conditions or treatments[14,15], which is essential for understanding cellular processes at a molecular level.

However, clustering SMLM data presents several challenges. Inherent localization noise, such as false positive identifications, can obscure true molecular patterns. Molecule undercounting and

[1]Department of Physics, University of Gothenburg, Origovägen 6B, SE-41296 Gothenburg, Sweden. [2]Facultat de Ciències, Tecnologia i Enginyeries, Universitat de Vic—Universitat Central de Catalunya (UVic-UCC), C. de la Laura, 13, 08500 Vic, Spain. [3]Bioinformatics and Bioimaging, Institut de Recerca i Innovació en Ciències de la Vida i de la Salut a la Catalunya Central (IRIS-CC), 08500 Vic, Spain. [4]Science for Life Laboratory, Department of Physics, University of Gothenburg, Origovägen 6B, SE-41296 Gothenburg, Sweden. ✉e-mail: giovanni.volpe@physics.gu.se; carlo.manzo@uvic.cat

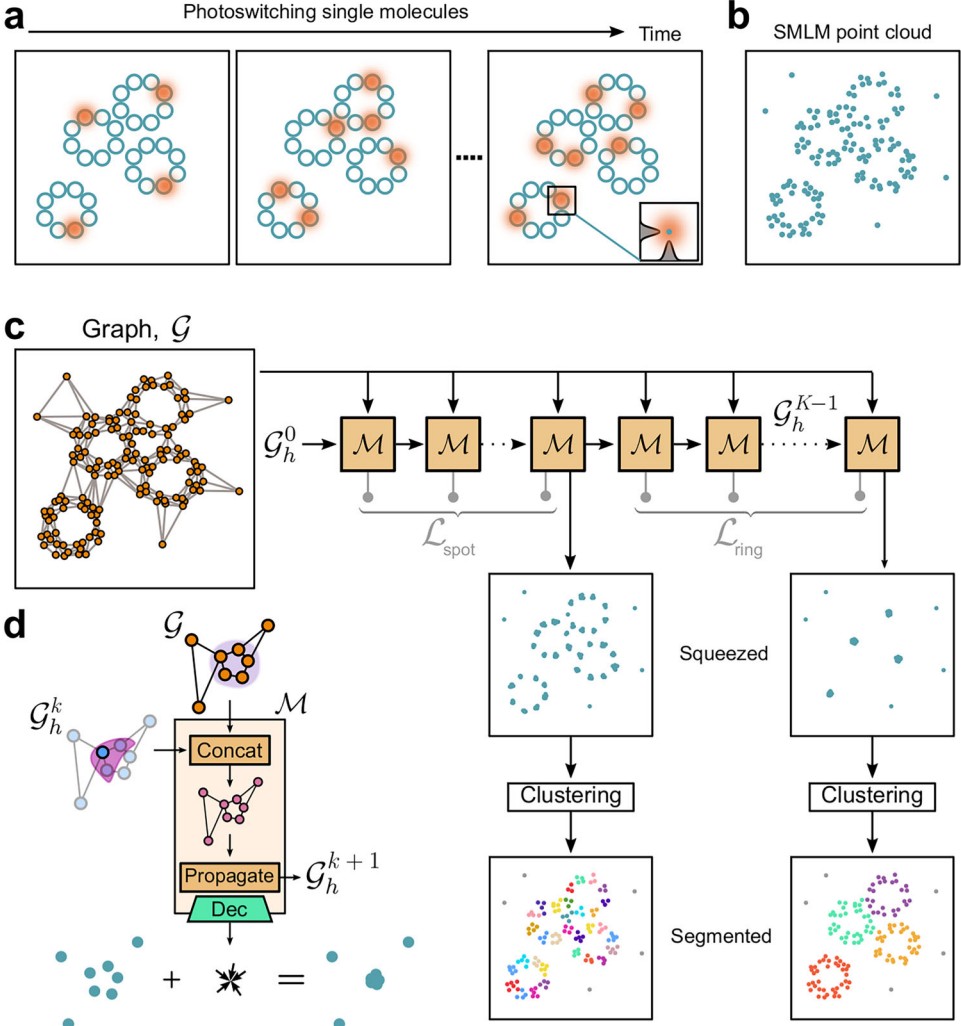

**Fig. 1 | Overview of the MIRO-based clustering workflow. a** Illustration of the SMLM image acquisition process for molecules organized in ring-shaped clusters. Molecules appear stochastically as bright fluorescence spots in different frames. The fluorescence intensity profile (inset) is used to precisely determine the molecular centroids. **b** The cumulative localizations from all frames are then combined to generate the experimental point cloud. **c** The molecular localizations are represented as a graph that is encoded in a latent representation $\mathcal{G}$, combined with a hidden graph $\mathcal{G}_h^k$, and recurrently processed through the MIRO block, $\mathcal{M}$. The hidden node features are used to minimize the loss functions (e.g., $\mathcal{L}_{spot}$ and $\mathcal{L}_{ring}$) calculated at each step, providing flexibility to use different ground truths across steps and thus enabling the network to collapse structures at various scales. Finally, the collapsed localizations are postprocessed through a conventional clustering algorithm to group those within the same structure. **d** The core operations of the MIRO block include the concatenation of the input graph $\mathcal{G}$ with the hidden graph $\mathcal{G}_h^k$. The input graph provides semantic information (e.g., the position of localization forming the same cluster, represented by the shaded circle). In contrast, the hidden graph $\mathcal{G}_h^k$ captures relational information between adjacent localizations (represented by the purple area). Information is propagated to generate an updated hidden graph $\mathcal{G}_h^{k+1}$, which is passed together with $\mathcal{G}$ to the next iteration of the MIRO block. A decoder produces displacement vectors from hidden node features that, when summed with the localization coordinates, shift localizations belonging to the same cluster toward a common center, leaving background localizations unaltered.

overcounting, where the same molecule is either not detected or detected multiple times due to photophysical effects, can distort the true distribution of molecules[9]. Molecular structures can be closely spaced and even overlapping, resulting in a high density of localizations that complicates the identification of distinct clusters.

Several algorithms have been specifically proposed for this task[16–22] and their performance has been recently assessed[23]. Among the methods evaluated in ref. 23, density-based spatial clustering of applications with noise (DBSCAN)[24], one of the most popular algorithms used for SMLM data, has been shown to be adaptable to diverse clustering conditions and to provide close-to-optimal performance, comparable to those obtained with the topological mode analysis tool (ToMATo)[21] and kernel density estimation (KDE). DBSCAN was also found to be the most robust to multiple blinking[23]. More recently, DBSCAN has been shown to achieve significantly higher scores than

HDBSCAN[25] and OPTICS[26] across different cluster types[27]. However, DBSCAN's performance is highly dependent on the choice of its two parameters: the maximum distance between two points for them to be considered as part of the same cluster ($\varepsilon$); and the minimum number of points that must be within a point's $\varepsilon$-neighborhood for that point to be considered a core point and thus form a cluster (minPts). These parameters determine what constitutes a cluster and what constitutes noise. Their choice can significantly affect the resulting clusters, and they require careful dataset-specific settings based on heuristic rules[18,28] or further analysis[27,29].

Moreover, biological clusters corresponding to supramolecular organizations often have non-trivial shapes, such as focal adhesions[12] or nuclear pore complexes[13]. These structures pose additional challenges due to their irregular or complex geometries. Traditional clustering methods work well with symmetric, simply connected, or

convex shapes, but often fail with non-symmetric, irregular, or highly complex distributions. These limitations highlight the necessity for improved clustering techniques that can extract meaningful information from SMLM data, ensuring accurate and reliable insights into the molecular architecture of biological samples.

In this paper, we introduce a novel supervised approach to enhance the versatility of clustering algorithms. Our method, MIRO (Multifunctional Integration through Relational Optimization), employs a few-shot (or one-shot) geometric deep learning framework based on recurrent graph neural networks (rGNNs) to learn a transformation that squeezes elements of complex point clouds around a common center (Fig. 1b, c). To achieve this, MIRO assumes that clusters' general structure and spatial relationships are preserved within a given dataset and uses relational information to make complex data more suitable for conventional clustering techniques. In this way, MIRO transforms the point clouds so that methods for complete clustering (i.e., assigning every localization to a specific cluster or to the non-clustered group[23]) can achieve enhanced performance, as we demonstrate for DBSCAN on a wide range of datasets with varied cluster shape and symmetry. By enhancing the spatial separation between localizations in adjacent clusters, as well as between clustered and background localizations, MIRO inherently simplifies the parameter selection for DBSCAN and similar methods. Additionally, the recurrent structure of MIRO facilitates a multifunctional representation framework, enabling the simultaneous handling of heterogeneous analysis tasks, such as multiscale clustering, clustering of differently shaped structures, and node-level classification. Unlike traditional pipelines that treat these tasks separately or require manual tuning, MIRO learns a flexible representation that supports diverse objectives in a unified and scalable manner. This multifunctional capability significantly expands the range of biologically relevant insights that can be extracted from a single experiment.

Following a recent benchmark study[23], we provide a comprehensive evaluation of MIRO's performance across various SMLM experimental scenarios, demonstrating its transformative potential for clustering applications. Furthermore, our analysis extends beyond this benchmark, showing that MIRO significantly improves clustering performance in complex and irregular data scenarios.

## Results

### MIRO workflow

MIRO uses relational information to transform point clouds to bring together points that belong to the same cluster. It achieves this by using a rGNN, which incorporates several innovative aspects in the architecture, operational mechanisms, and training process, as described here. A detailed description is provided in the "Methods".

MIRO is built on an rGNN architecture[30]. The input to the neural network is a graph representation of individual molecular localizations derived from SMLM experiments[31]. As shown in Fig. 1a, these localizations are obtained from multiple fluorescence images of the same field of view (FOV), with each image capturing a sparse number of simultaneously emitting fluorophores. Importantly, fluorophores' emission is stochastic; therefore, a given fluorophore can be detected in multiple frames or not at all. The images are processed to extract the centroid positions of bright features corresponding to molecular localizations. These positions are then drift-corrected and filtered to remove low-quality localizations. Additionally, localizations that are too close together within the same field are discarded, while those that appear in consecutive frames are merged to ensure an accurate representation of distinct molecules.

In the graph representation, each node is associated with a single molecular localization, while edges capture spatial relationships between nodes within the point cloud (Fig. 1b, c). Edges are derived from a Delaunay triangulation and filtered according to a distance threshold to prevent spurious connections in low-density regions.

Absolute positional information is not directly used as a node feature but solely to define connectivity. Instead, node features are encoded using Laplacian positional embeddings[32], while edge features include the Euclidean distance and a direction vector.

To strengthen the ability to capture complex spatial relationships, the graph is encoded into a higher-level representation $\mathcal{G}$ through a learnable dense layer followed by ReLU activation. The latter serves as the input of a sequence of recurrent steps, were the same MIRO block $\mathcal{M}$ is repeatedly applied, as shown in Fig. 1c. We emphasize that MIRO uses a single-layer architecture. As a result, increasing the number of recurrent steps does not affect the number of learnable parameters. The number of recurrent steps defines the size of the receptive field and therefore needs to be adapted to the density of the point cloud and the complexity of the clustering problem, as discussed in Number of recurrent steps: influence on performance and oversmoothing.

The operations of an MIRO block are schematically illustrated in Fig. 1d. At each recurrent step, the graph $\mathcal{G}$ is concatenated with a "hidden" graph $\mathcal{G}_h^k$ having the same structure and with node and edge features initialized to zeros. Similar to the hidden state of a recurrent neural network, $\mathcal{G}_h^k$ represents the hidden state of the system and characterizes the underlying processes being modeled, capturing relational information between nearby localizations. Information is propagated to generate an updated hidden graph $\mathcal{G}_h^{k+1}$ that is passed to the next step, together with the unmodified $\mathcal{G}$. In contrast to typical message passing schemes[30,33,34], MIRO omits the concatenation of node features with aggregated messages. Instead, hidden node features are updated solely based on hidden edge features (i.e., the messages) to emphasize the immediate structural context of each node. The hidden node features are further decoded through a learnable dense layer to provide, for each molecular localization, a displacement vector in Cartesian space. These displacements are calculated to minimize a loss function $\mathcal{L}$ (see MIRO loss function) that aims to shift localizations belonging to the same cluster toward a common center, while leaving background localization unaltered.

To ensure a meaningful hidden representation and prevent vanishing gradients, at each iteration within an epoch, the loss is further averaged across all recurrent steps[30], as schematically shown in Fig. 1c. This approach imposes intermediate corrections to the displacement vectors, helping to maintain the clusters' structural integrity throughout the recurrent steps. This method also allows for different steps in the process to have different ground truths, enabling the network to learn and adapt to multiscale features—like the circular clusters ($\mathcal{L}_{spot}$) and the ring structures ($\mathcal{L}_{ring}$) shown in the example of Fig. 1c. Such multiscale training enhances MIRO's ability to handle varying cluster sizes, shapes, and densities within the same dataset, further improving its robustness and accuracy in clustering complex biological data.

Notably, MIRO's training can be effectively performed using a single or a few representative clusters (see MIRO training and augmentations). This approach uses the weak conservation of shape and organization within molecular clusters to boost clustering accuracy. By employing a series of augmentations, the algorithm learns to generalize across a given scenario, enabling robust performance even when trained on minimal data.

### MIRO enhances DBSCAN performance

To demonstrate the benefits of using MIRO, we first applied it to simulated datasets, as illustrated in Fig. 2. MIRO is designed as a pre-processing step to enhance the performance of subsequent clustering methods. To assess the performance gains introduced by MIRO, we compared the results of DBSCAN both with and without MIRO pre-processing. We selected DBSCAN for this comparison due to its top performance in benchmark studies[23,27] and its widespread use in the literature[28].

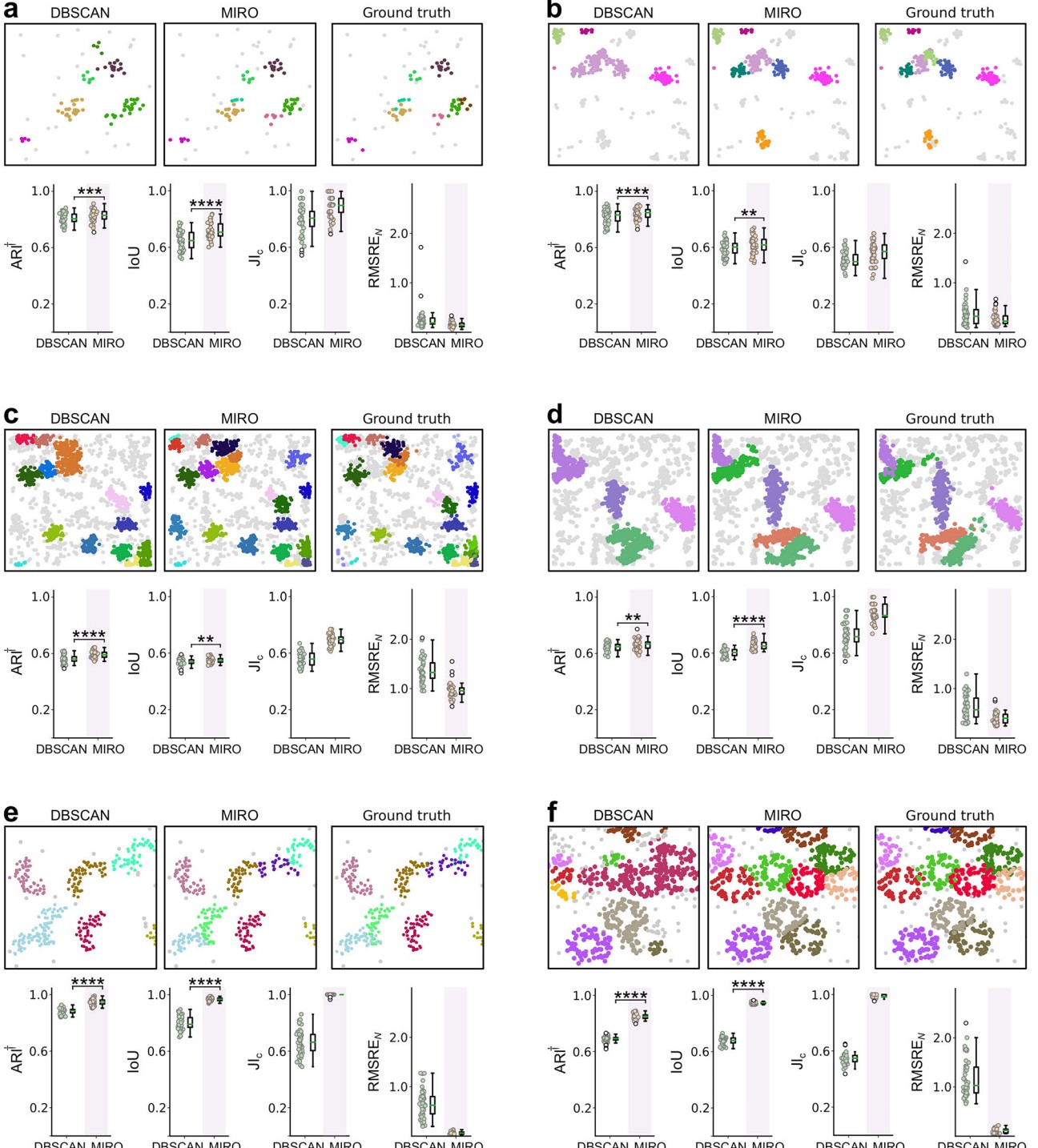

**Fig. 2 | MIRO clustering performance on simulated datasets.** Each panel represents results obtained for one dataset: **a** Scenarios 8; **b** Scenarios 8 with blinking; **c** Scenarios 5 with blinking; **d** Scenarios 6 with blinking; **e** C-shaped clusters; and **f** ring-shaped clusters. Within each panel, the upper row shows an exemplary FOVs with localizations analyzed by DBSCAN alone (left), DBSCAN with MIRO pre-processing (middle), and the ground truth (right). Localizations are color-coded according to their assigned clusters. The bottom row presents scatter plots of the robust variant of the Adjusted Rand Index (ARI†), the intersection over union (IoU), the Jaccard Index for cluster detection (JI_c), and the root mean squared relative

error in the number of localizations per cluster (RMSRE_N) calculated over 47 (50 for **e** and **f**) different simulations (filled circles), together with their box-and-whisker plot. The central line represents the median, the box edges represent the first and third quartiles, the whiskers extend to the most extreme data points within 1.5 times the interquartile range, and outliers are shown as empty circles. Statistical significance was assessed through a paired one-sided Wilcoxon test. The number of stars represents the level of statistical significance (*$p \leq 0.5$; **$p \leq 0.01$; ***$p \leq 0.001$; ****$p \leq 0.0001$). Exact $p$ values for all statistical comparisons are provided in the accompanying Source Data file. Source data are provided as a Source Data file.

For the benchmark datasets, we used the DBSCAN parameters provided in ref. 23. For MIRO-preprocessed data and other datasets, clustering parameters were optimized using an automated procedure based on the Optuna Python library[35], guided by metric-based performance scores. These parameters were consistently applied across all experiments within the same scenario and are summarized in Supplementary Tables 1 and 2. Please refer to DBSCAN parameter selection for a discussion of the parameter choice criteria.

Clustering performance was evaluated using various metrics. The benchmark study[23] employed the adjusted Rand index (ARI)[36] to evaluate cluster membership and the intersection over union (IoU) to measure the overlap of clusters defined by their convex hulls. However, ARI is known to be highly sensitive to cluster size imbalances[37,38], a common issue in SMLM data where non-clustered molecules are often treated as an additional "background" cluster. To handle the effect of imbalance, we employed alternative metrics better suited for these scenarios, including a robust variant of ARI (ARI†)[38], adjusted mutual information (AMI)[37,39], and ARI calculated excluding non-clustered localizations (ARI$_c$)[22]. Further details on these metrics can be found in Metrics for performance evaluation.

In addition to these metrics, we used cluster-level metrics such as the Jaccard Index for cluster detection (JI$_c$), the root mean squared relative error in the number of localizations per cluster (RMSRE$_N$), and the root mean squared error in cluster centroid position (RMSE$_{x,y}$).

In our evaluation, MIRO consistently enhances the performance of DBSCAN across all tested scenarios, as shown in Supplementary Tables 3 and 4. While these results are based on few-shot training (three FOVs, comprising 60 to 300 clusters), we also demonstrate that comparable performance can be achieved with single-shot training, as shown in Supplementary Table 5 for representative scenarios.

First, we discuss MIRO's performance on selected datasets from the benchmark study[23], characterized by different cluster density, size, and shape (Table 1 and Fig. 2a–c). For instance, in Scenario 8 (small symmetrical clusters with two different densities), while the scatter and box plots in Fig. 2a show that MIRO only slightly improves the performance of DBSCAN, this improvement is statistically significant. This scenario represents a case where the performance of DBSCAN is close-to-optimal, therefore, it is not surprising that MIRO only makes a small difference. Specifically, MIRO achieves a medium effect size for ARI† (Cohen's $d = 0.5$; paired one-sided Wilcoxon test $W = 852$, $n = 47$, $p = 9.3 \times 10^{-4}$) and a large effect size for IoU (Cohen's $d = 0.89$; $W = 993$, $n = 47$, $p = 5.3 \times 10^{-7}$), with this improvement being most pronounced in cluster-level metrics such as JI$_c$ (Cohen's $d = 0.92$; $W = 730$, $n = 47$, $p = 1.0 \times 10^{-6}$) and RMSRE$_N$ (Cohen's $d = 0.48$; $W = 21$, $n = 47$, $p = 3.2 \times 10^{-12}$).

As expected, the advantage of using MIRO becomes more evident in more challenging conditions. In Scenario 8 with blinking (Fig. 2b), the increased number of localizations due to molecular overcounting introduces more heterogeneity into the data, but MIRO effectively mitigates this effect and significantly improves DBSCAN's performance (Cohen's $d = 0.75$; $W = 956$, $n = 47$, $p = 5.9 \times 10^{-6}$ for ARI†). MIRO further demonstrates its capability to handle additional complexities when, in addition to blinking, the number of clusters is increased, as in Scenario 5 (Cohen's $d = 1.51$; $W = 1114$, $n = 47$, $p = 7.8 \times 10^{-13}$ for ARI†, Fig. 2c).

To further highlight MIRO's ability in managing complex cluster geometries, we evaluated its performance under three additional conditions. First, we examined Scenario 6 with blinking, which includes elliptically shaped clusters. Additionally, we simulated data with C-shaped and ring-shaped clusters (Fig. 2d–f). In these scenarios, MIRO produces a marked enhancement in DBSCAN's performance by consistently transforming elongated and non-convex shapes into well-defined, compact clusters for the further application of DBSCAN.

We further demonstrate that MIRO outperforms recent supervised methods not included in the benchmark, such as an implementation of the GNN-based framework proposed in ref. 22 (MAGIK-S), as shown in Supplementary Table 6. Details of the implementation are provided in Comparison with a supervised graph-based clustering framework.

## Simultaneous clustering and classification of different shapes

MIRO offers the capability of simultaneously handling diverse structural patterns by compressing localizations from different cluster shapes into a uniform representation. This capability enables effective

**Table 1 | Summary of clustering metrics for different scenarios and methods**

| Scenario | Method | ARI† | IoU | JI$_c$ | RMSRE$_N$ | RMSE$_{x,y}$ | AMI | ARI$_c$ | ARI |
|---|---|---|---|---|---|---|---|---|---|
| Scenario 8 | MIRO | 0.83±0.04 | 0.72±0.06 | 0.88±0.08 | 0.16±0.06 | 2.3±0.3 | 0.85±0.03 | 0.92±0.04 | 0.83±0.03 |
| | DBSCAN | 0.81±0.04 | 0.65±0.06 | 0.80±0.11 | 0.3±0.2 | 2.6±0.3 | 0.83±0.03 | 0.90±0.07 | 0.80±0.04 |
| Scenario 8 blinking | MIRO | 0.84±0.04 | 0.62±0.06 | 0.57±0.07 | 0.28±0.13 | 2.6±0.4 | 0.77±0.05 | 0.81±0.08 | 0.65±0.08 |
| | DBSCAN | 0.82±0.05 | 0.59±0.05 | 0.52±0.05 | 0.4±0.2 | 2.9±0.6 | 0.76±0.04 | 0.74±0.08 | 0.64±0.06 |
| Scenario 5 blinking | MIRO | 0.59±0.02 | 0.56±0.02 | 0.70±0.04 | 0.78±0.13 | 3.71±0.14 | 0.62±0.02 | 0.50±0.05 | 0.37±0.03 |
| | DBSCAN | 0.56±0.03 | 0.54±0.03 | 0.56±0.05 | 1.3±0.2 | 3.92±0.15 | 0.60±0.02 | 0.28±0.05 | 0.42±0.03 |
| Scenario 6 blinking | MIRO | 0.66±0.04 | 0.66±0.03 | 0.89±0.06 | 0.40±0.12 | 4.3±0.5 | 0.74±0.02 | 0.84±0.05 | 0.65±0.03 |
| | DBSCAN | 0.65±0.03 | 0.61±0.03 | 0.72±0.09 | 0.6±0.2 | 4.8±0.5 | 0.70±0.02 | 0.65±0.06 | 0.66±0.03 |
| C-shaped | MIRO | 0.95±0.02 | 0.968±0.011 | 0.999±0.005 | 0.06±0.03 | 0.27±0.06 | 0.967±0.011 | 0.95±0.02 | 0.94±0.02 |
| | DBSCAN | 0.88±0.02 | 0.80±0.05 | 0.67±0.09 | 0.6±0.3 | 0.68±0.11 | 0.90±0.02 | 0.72±0.09 | 0.71±0.08 |
| Rings | MIRO | 0.85±0.02 | 0.947±0.006 | 0.990±0.012 | 0.11±0.05 | 0.048±0.004 | 0.909±0.010 | 0.86±0.02 | 0.82±0.02 |
| | DBSCAN | 0.69±0.02 | 0.68±0.02 | 0.55±0.04 | 1.2±0.4 | 0.151±0.005 | 0.73±0.02 | 0.34±0.05 | 0.33±0.04 |

Data represent mean ± standard deviation calculated over 47 fields of view (50 for C-shaped and rings scenarios). Complete results for all scenarios are presented in Supplementary Tables 3 and 4, corresponding to datasets without and with blinking, respectively.

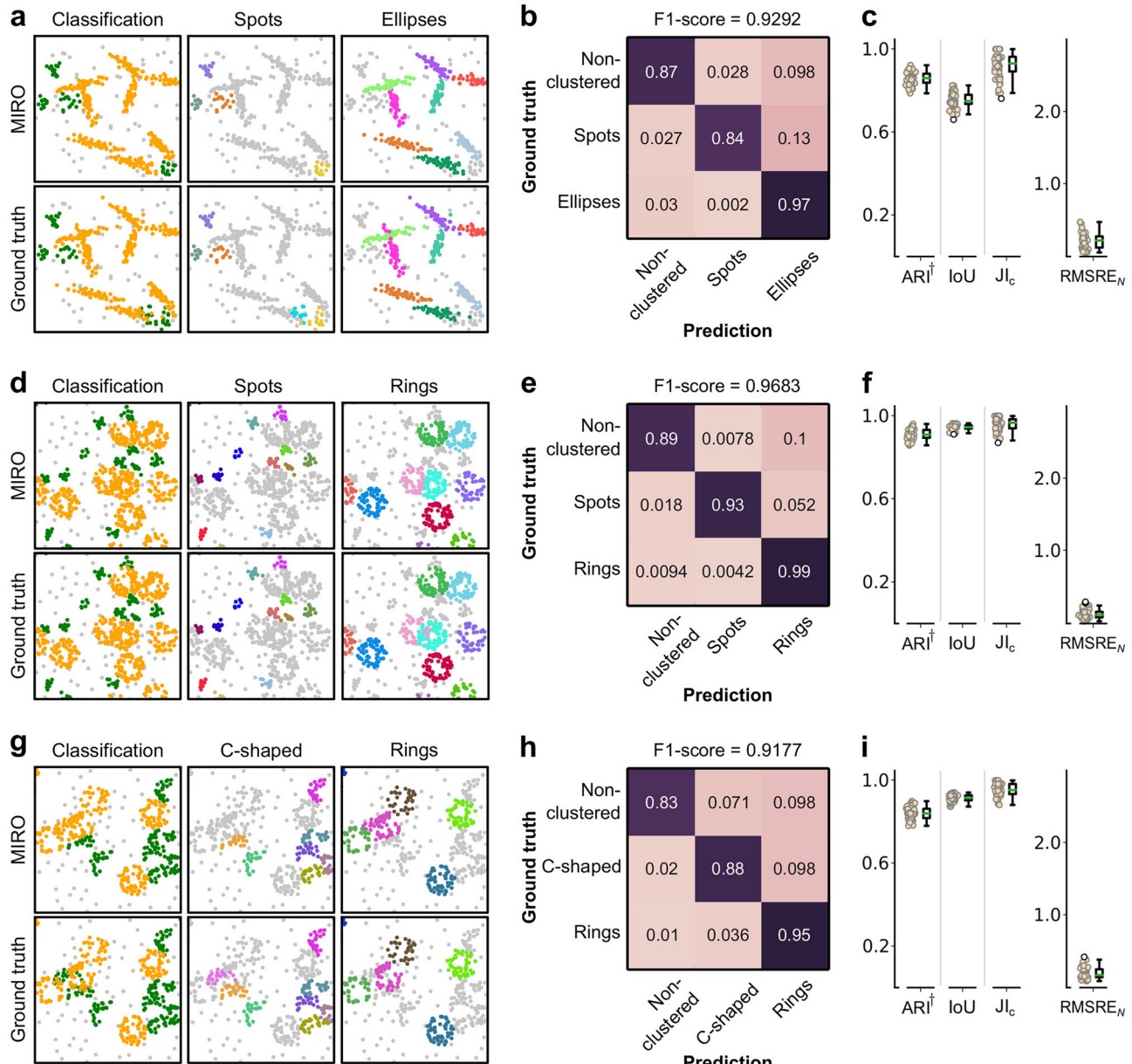

**Fig. 3 | MIRO's simultaneous clustering and classification of different shapes.** Results from simulations involving three distinct mixtures of shapes: **a**–**c** spots and ellipses, **d**–**f** spots and rings, and **g**–**i** C-shaped clusters and rings. **a**, **d**, **g** Exemplary fields of view with the mixtures analyzed using DBSCAN with MIRO preprocessing (top) alongside the ground truth (bottom). Localizations are color-coded. In the left column, different colors correspond to different shapes, while non-clustered localizations are shown in gray. In the middle and right columns, localizations forming clusters of specific shapes are color-coded based on their assigned clusters, with other shapes and non-clustered localizations depicted in gray. **b**, **e**, **h** Confusion matrices with the classification accuracy for different structural configurations. The rows represent the true classes, and the columns represent the predicted classes, with F1-scores indicated to assess the overall classification performance. **c**, **f**, **i** Box-and-whisker plots of the robust variant of the Adjusted Rand Index (ARI†), the intersection over union (IoU), the Jaccard Index for cluster detection (JI$_c$), and the root mean squared relative error in the number of localizations per cluster (RMSRE$_N$), calculated across 50 simulations (filled circles). The central line in each boxplot represents the median, the box edges correspond to the first and third quartiles, the whiskers extend to the most extreme data points within 1.5 times the interquartile range, and outliers are shown as empty circles. Source data are provided as a Source Data file.

clustering using a single set of parameters across different shapes when applied to algorithms like DBSCAN. The unified representation simplifies parameter tuning and enhances clustering performance. However, this transformation can also lead to challenges in subsequent classification, as the uniform collapse of different shapes may obscure their unique features.

However, while transforming various structures into compact forms, MIRO can generate additional output features at the node level that can be used, e.g., for simultaneous cluster shape classification. This dual capability is essential for, e.g., distinguishing among various molecular assemblies within the same biological environment, each exhibiting unique organizational patterns and functional roles.

To evaluate MIRO's ability to simultaneously cluster and classify different structures, we generated simulated datasets comprising mixtures of circular, elliptical, C-shaped, and ring-shaped clusters (Fig. 3). Each cluster type represents a distinct molecular assembly, characterized by unique spatial properties. MIRO effectively learns to capture these features at the node level. While clustering ensures accurate separation of structures, taking the mode of node-level class

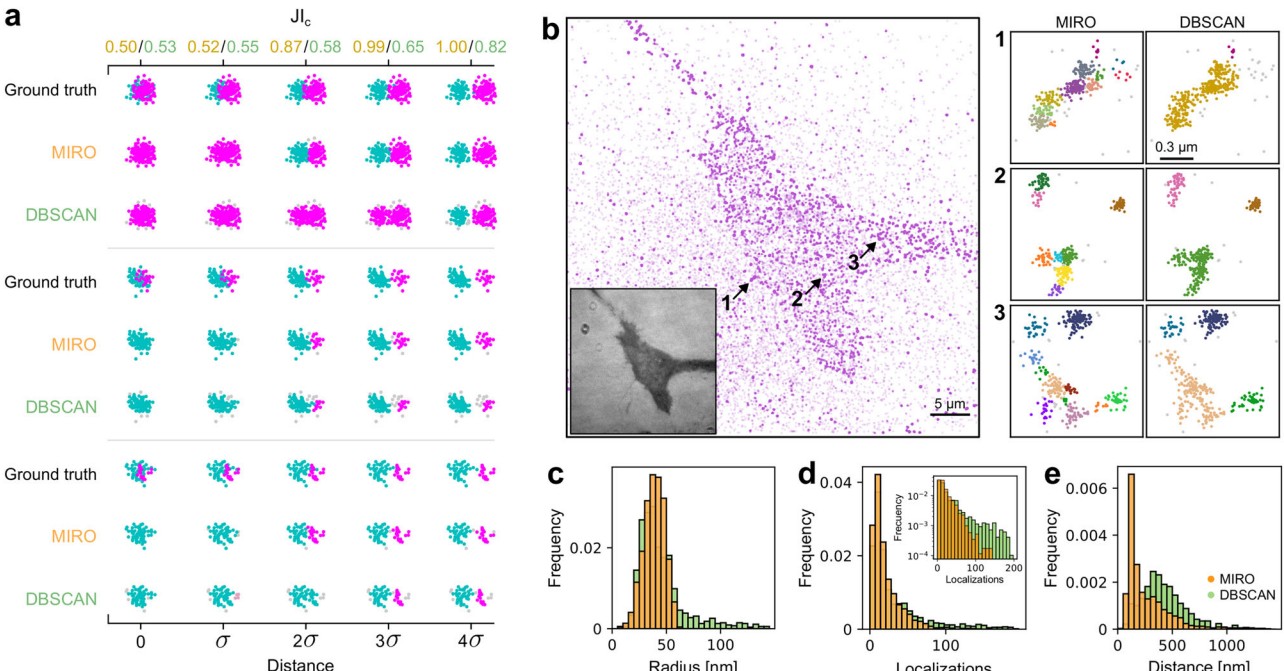

**Fig. 4 | MIRO improves the quantification of dense and heterogeneous clusters.** **a** Performance comparison of MIRO and DBSCAN in resolving cluster pairs located at varying distances relative to their radius $\sigma$. The panel illustrates three examples with different numbers of localizations. The Jaccard Index for cluster detection ($JI_c$), calculated as a function of distance, demonstrates the superior performance achieved using MIRO over DBSCAN alone. **b** Localization map obtained from a dSTORM image of integrin $\alpha5\beta1$ in HeLa cells, analyzed using MIRO. Clustered localizations are represented by opaque symbols, while semi-transparent symbols represent non-clustered localizations. The numbered panels on the right are zoomed-in views of the regions indicated by the arrows, with different colors representing different clusters identified by MIRO (left column), whereas DBSCAN merges adjacent clusters (right column). Scale bar 5 μm. (Lower inset) Reflection interference contrast image of the cell, darker regions correspond to the membrane adhesion area. Quantification of the clustering obtained by MIRO (orange) and DBSCAN (green): **c** histogram of cluster radius, **d** number of localizations per cluster (logarithmic $y$-scale in the inset), and **e** the nearest neighbor distance between clusters. Source data are provided as a Source Data file.

predictions within each cluster allows for reliable identification of the corresponding structural type in heterogeneous datasets.

Figure 3 illustrates the results for three distinct mixtures: spots and ellipses (Fig. 3a–c), spots and rings (Fig. 3d–f), and C-shaped clusters and rings (Fig. 3g–i). Overall, the results demonstrate that MIRO's preprocessing effectively distinguishes between different shapes and accurately assigns localizations to their respective clusters. This enhanced performance is evident in both the confusion matrices (Fig. 3b, e, h), which show higher classification accuracy across all shape combinations, and the clustering metrics (Fig. 3c, f, i). Notably, the clustering metrics indicate that, in several instances, the performance is similar to those obtained for a single shape. This is particularly remarkable considering that no restrictions were imposed on cluster overlap; clusters of different shapes could overlap or be arranged in ways that mimic other shapes, such as aligned spots forming an ellipse or facing C-shapes resembling a ring.

## Detecting heterogeneous and dense clusters

In SMLM, fluorophore blinking often results in overcounting, where each molecule produces multiple localizations. This phenomenon creates artificial clusters with dimensions comparable to the localization precision[9]. Additionally, the natural aggregation of proteins at the nanoscale leads to the formation of structures known as nanoclusters[10], which further contributes to clustering.

Accurate clustering analysis is crucial for precisely quantifying the spatial distribution of these nanoclusters. This involves tasks such as determining nanocluster sizes and estimating protein copy numbers within each nanocluster, often in comparison to a reference sample[40]. High cluster density or supra-cluster organization exacerbates the challenge, as reduced inter-cluster distances and variable localization counts between adjacent clusters can lead to the underestimation of the number of clusters and the overestimation of cluster sizes and molecular content.

MIRO offers substantial improvements for analyzing adjacent clusters in SMLM data. We assessed MIRO's effectiveness by conducting quantitative tests as a function of the inter-cluster distance. We simulated pairs of clusters with similar sizes but containing different numbers of localizations, located at varying cluster-to-cluster distances. Localizations belonging to the same cluster were spatially arranged according to a 2D Gaussian distribution with width $\sigma$. The number of localizations per cluster was drawn from an exponential distribution. Clusters were spaced at various distances as a function of $\sigma$. We applied MIRO and DBSCAN to compare the methods' ability to resolve the clusters, as quantified by the Jaccard Index for cluster detection ($JI_c$). As demonstrated in Fig. 4a, at distances $\leq 2\sigma$, MIRO significantly improves clustering accuracy compared to DBSCAN, providing a more precise characterization of nanocluster spatial arrangements and thus improving their quantification.

Additionally, we applied MIRO to the quantification of molecular organization in experimental data. Using dSTORM images of integrin $\alpha5\beta1$ in HeLa cells, we studied receptor organization, which exhibits a spatial hierarchy with molecules arranged in nanoclusters[41] that can aggregate to form larger structures that build focal adhesions (FAs)[12,15]. MIRO processing of molecular localizations allowed for accurate identification of integrin nanoclusters, as shown in Fig. 4b. The cell area, corresponding to the dark region in the reflection interference contrast image (inset of Fig. 4b), reveals a high density of nanoclusters (opaque symbols). The zoomed-in regions 1–3 in Fig. 4b illustrate MIRO's ability to resolve close individual nanoclusters forming larger structures, whereas DBSCAN merges nearby clusters.

Thanks to the robust identification of the nanoclusters enabled by MIRO, it is then possible to precisely quantify nanocluster size (Fig. 4c), number of localization per nanocluster (Fig. 4d), and distance between nanoclusters (Fig. 4e), providing a more accurate and detailed understanding of molecular organization as compared to DBSCAN alone and underscoring MIRO's potential for high-resolution analysis of protein complexes in SMLM. Clusters retrieved by MIRO show a monodispersed distribution of radius, centered at ≈38 nm (Fig. 4c), and a distribution of the number of localizations per cluster with an exponential tail with an average of 17.8 (Fig. 4d), whereas DBSCAN shows spurious longer tails in both distributions, due to the merging of adjacent clusters. As a consequence, Fig. 4e shows that the nearest-neighbor distance between nanoclusters calculated on MIRO-processed data has a peak at ≈100 nm, reflecting cluster proximity that DBSCAN misses due to the merging of adjacent clusters.

### Multiscale clustering of nuclear pore complex

Molecular complexes often exhibit organization across multiple scales, with the nuclear pore complex (NPC) being a paradigmatic example. The NPC is a large molecular channel embedded in the nuclear envelope, regulating the transport of macromolecules between the nucleus and cytoplasm of eukaryotic cells. The NPC consists of more than 30 proteins and has a precise three-dimensional architecture. One of its key components, Nup96, is present in 32 copies per NPC, forming both a cytoplasmic ring and a nucleoplasmic ring. Each ring features 8 corners, with two Nup96 molecules at each corner. When imaged with SMLM, Nup96-labeled NPCs oriented parallel to the focal plane display an annular structure. Since the two rings are nearly aligned, the eightfold symmetry of the NPC is clearly observable and each of the eight corners thus appears as a small cluster of the localizations generated by four Nup96 molecules. Because of its regular arrangement, Nup96 endogenously tagged with commonly-used labels has been adopted as a reference protein for the quantitative optimization of super-resolution microscopy workflows[13].

The characterization of the nuclear pore complexes from SMLM imaging poses a challenge at two different scales: accurate segmentation of the ring structures and precise identification of the corners. Both tasks are typically tackled separately with ad hoc methods, which are often strongly dependent on algorithmic parameters. However, thanks to its sequential architecture, MIRO enables the simultaneous segmentation of rings and corners.

To demonstrate MIRO's ability to tackle these challenges simultaneously and quantitatively, we first relied on simulations. We generated synthetic localization maps with structures composed of small symmetrical clusters, each with a random number of localizations, arranged in rings with eightfold symmetry. As shown in Fig. 1, the MIRO architecture was trained to collapse localizations forming the spots and the ring-shaped clusters toward their respective centers. The results of the ensuing clustering, shown in Fig. 5a, b, demonstrate that MIRO can work simultaneously across multiple scales. Specifically, MIRO provides significant performance enhancements compared to DBSCAN at both the ring and spot scales, achieving better scores in all metrics.

To further validate MIRO's effectiveness in clustering across multiple scales, we applied it to experimental data obtained from SMLM imaging of Nup96-nMaple in fixed U2OS cells in 50 mM Tris in $D_2O$ from ref. 13. The localization map, shown in Fig. 5c, displays localizations color-coded by rings identified by MIRO, with non-clustered localizations represented in gray. This visualization highlights MIRO's capability to accurately segment ring structures and distinguish between clustered and non-clustered molecules, even in densely packed regions. Figure 5d provides a zoomed-in view of selected ring-like structures, with different colors representing distinct corners, underscoring MIRO's ability to resolve structural details at a finer scale. Note that some missed corners are expected due to the effective labeling efficiency of only 58%[13].

The quantitative analysis of the clustering results is presented in Fig. 5e, f. The histogram of the number of localizations per NPC in Fig. 5e shows very good agreement with the one obtained in the original article (Fig. 4g in ref. 13), where segmentation was performed using a specifically designed algorithm involving multiple filtering and thresholding of reconstructed super-resolution images. Similarly, Fig. 5f presents a histogram of the number of localizations per spot, revealing an exponential distribution with an average of 12.28 localizations per spot. Considering that each corner hosts 4 Nup96 proteins, this result corresponds to approximately 3 localizations per protein, closely aligning with the estimation performed in the original article (2.8 localizations, Table 1 in ref. 13). These results highlight the accuracy and reliability of MIRO in multiscale real-data applications. In contrast, DBSCAN struggles in this scenario. As shown in Supplementary Fig. 1, identifying optimal DBSCAN parameters is non-trivial, and the algorithm often produces fragmented clusters, resulting in an artificial peak of small-sized clusters in the size distribution. Overall, these findings underscore the advantage of MIRO not only over traditional clustering approaches like DBSCAN, but also compared to the task-specific, multi-step analysis pipeline originally proposed for this dataset[13].

## Discussion

MIRO represents a significant contribution to the clustering of SMLM localizations through the application of rGNNs.

Preprocessing SMLM datasets with MIRO enhances the performance of algorithms for complete clustering. Accurate clustering enables the quantitative assessments of spatial organization within cellular environments, through the precise estimation of quantities such as cluster size, protein copy number, and inter-cluster distances, leading to deeper insights into biological phenomena[12,14,15].

The integration of MIRO allows for simultaneous clustering and classification of various structural patterns within a single dataset. To the best of our knowledge, this feature is not offered by any of the previous methods. Moreover, MIRO operates in both multiclass and multiscale modalities, with the multiscale approach being particularly advantageous for nested structures, such as NPCs[13]. It is important to note that the same MIRO block is used to compress structures of different sizes and shapes in a single forward pass; thus, the hidden representation inherently learns hierarchies and scales within the data.

MIRO advances data-driven analysis in SMLM by offering expanded functionality beyond existing approaches. Early machine learning methods, such as those based on recurrent neural networks, were limited to binary classification, distinguishing only between clustered and non-clustered localizations[20]. More recently, graph neural networks have been used to cluster SMLM data with simple symmetric shapes, but these models do not support the classification of structural types[22]. SEMORE[42] introduces a different strategy by applying machine learning for morphological fingerprinting of clusters obtained from density-based clustering. In contrast, MIRO focuses on learning robust spatial representations, enabling both improved clustering and simultaneous structural classification. As such, MIRO could serve as a valuable preprocessing step that complements methods like SEMORE, potentially improving the quality of the point clouds that SEMORE subsequently analyzes.

Although MIRO is trained in a supervised manner, we note that supervision alone does not guarantee improved clustering results—as shown in the benchmark study, where the supervised method CAML[20] does not outperform the best-performing unsupervised alternatives. MIRO's superior performance stems not simply from supervision, but also from its tailored architecture and inductive biases, which enhance representation quality and generalization even with limited training data.

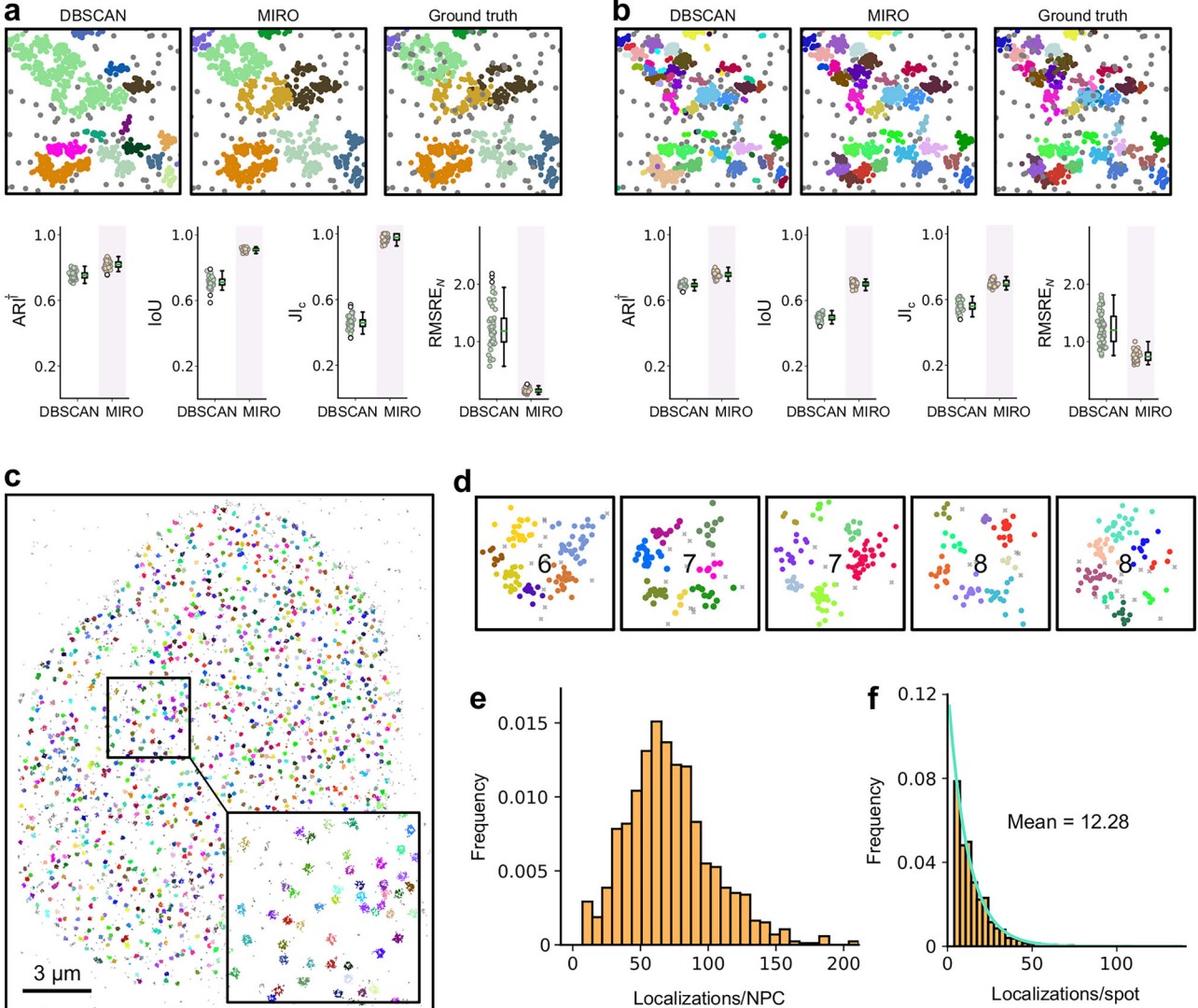

**Fig. 5 | MIRO allows simultaneous multiscale clustering.** Results obtained for the multiscale clustering of **a** rings and **b** spots within the same structure. The upper row shows an exemplary FOV with localizations analyzed by DBSCAN alone (left), DBSCAN with MIRO preprocessing (middle), and the ground truth (right). Localizations are color-coded according to their assigned clusters. The bottom row presents scatter plots of the robust variant of the Adjusted Rand Index (ARI†), the intersection over union (IoU), the Jaccard Index for cluster detection (JI$_c$), and the root mean squared relative error in the number of localizations per cluster (RMSRE$_N$) calculated over 50 different simulations (circles), together with their box-and-whisker plot. The central line represents the median, the box edges represent the first and third quartiles, the whiskers extend to the most extreme data points within 1.5 times the interquartile range, and outliers are shown as empty circles. **c** Localization map obtained from a STORM image from ref. 13. Localizations are colored according to the identified clusters, non-clustered localizations are shown in gray. The inset shows a zoomed-in view of the boxed region. Scale bar 3 μm. **d** Examples of corner structures identified by MIRO within ring-like structures. Localizations are colored according to the identified corner and non-clustered localizations are indicated by gray crosses. The numbers indicate the number of corners identified by the algorithm. Quantification of the clustering results at the two scales with the histogram of the number of localization per nuclear pore complex (**e**) and localizations per spot (**f**). The green line in **f** corresponds to an exponential fit, retrieving an average number of 12.28 localization per spot. Source data are provided as a Source Data file.

This point is further reinforced by our comparison with MAGIK-S (see Supplementary Table 6), an implementation of a recent supervised method based on graph neural networks[22]. While both MIRO and MAGIK-S incorporate relational inductive biases through the use of graph-based models, the two architectures are designed to tackle different tasks. MAGIK-S is structured as a two-step pipeline involving node-level and edge-level classification, ultimately relying on community detection to form clusters. This approach requires a more complex optimization process, longer training times, and a greater amount of labeled data. In contrast, MIRO is designed specifically to enhance latent representations through recurrent graph operations, allowing for efficient single-shot or few-shot learning and fast generalization across scenarios.

Moreover, the architectural biases in MIRO are fundamentally different from those in MAGIK-S. MIRO explicitly decouples topological refinement from semantic input by maintaining access to the original graph structure across recurrent steps, which enables it to build robust, scale-adaptive representations. This design leads to improved clustering performance with significantly lower computational and data requirements.

MIRO transforms the parameter space in a way that makes the precise selection of parameters of DBSCAN less critical, thus improving the robustness and reliability of the clustering results. This is particularly important because the choice of parameters in DBSCAN can significantly affect the clustering outcome[23] and its unbiased selection requires the application of ad hoc procedure or algorithms[28,29].

In addition, MIRO's single- or few-shot learning capability allows it to generalize across scenarios with minimal training, making it highly efficient and versatile. As a result, MIRO is particularly well-suited when labeled data is limited or expensive to obtain. Its efficiency in learning from a small number of samples also translates to faster training times and reduced computational resources, further enhancing its practicality and appeal for real-world use cases.

From an architectural point of view, MIRO tackles several technical challenges and introduces an innovative scheme for the application of rGNNs in the analysis of point clouds.

A key challenge addressed by MIRO is the analysis of high-density localization maps, which requires a broad receptive field to capture spatial relationships within dense point clouds. In conventional message-passing architectures, achieving this typically demands deeper networks, resulting in increased computational cost, memory usage, and a higher risk of oversmoothing[43]. MIRO overcomes these limitations by employing a recurrent architecture that progressively expands the receptive field without increasing the number of trainable parameters.

MIRO adopts a non-conventional approach to message passing[30], aiming to progressively refine topological information, captured in the hidden state, while maintaining access to the semantic information encoded in the original graph. To achieve this, MIRO's update block is specifically designed to focus on the local structural context: hidden node features are updated exclusively from the aggregated messages (i.e., hidden edge features), without concatenating previous node features. Meanwhile, the unmodified input graph is passed through each recurrent step, preserving the original semantic features and ensuring they remain accessible throughout the computation. This architectural choice enables MIRO to disentangle structural and semantic cues, leading to more interpretable representations that are particularly well-suited for identifying spatially coherent clusters.

While MIRO offers significant advantages, it is not without limitations. One fundamental challenge of MIRO, and all clustering methods, is accurately identifying and separating structures overlapping with either noise or other structures. Future improvements in this sense will be crucial for advancing clustering methods in complex biological datasets. Help in this sense might come from extending the node features. Node features in MIRO can encompass a wide range of attributes, providing flexibility in data representation. While our current implementation does not utilize temporal information, incorporating such data in an embedded form could enrich the model's performance by accounting for photophysical effects[42,44–46].

Beyond SMLM, MIRO's core capabilities make it a promising tool for a range of scientific domains. In neuroscience, for example, MIRO could help delineate complex neural circuits from sparse imaging data, advancing our understanding of brain connectivity[47]. Similarly, in environmental science, it could aid in uncovering spatial patterns in ecological datasets, such as species distributions or pollution gradients[48], contributing to more data-driven environmental monitoring and modeling.

## Methods

### MIRO graph representation

The input to MIRO is a graph representation[34] of an SMLM point cloud (Fig. 1a–c). In this graph, nodes (V) represent individual molecular localizations and edges (E) capture the spatial relationships within the point cloud derived from a Delaunay triangulation. To ensure that only meaningful, local spatial relationships are retained, the edges are filtered based on a distance threshold $\delta$ selected according to the local density of the point cloud (Supplementary Table 7).

Nodes are described by a set of features $\mathbf{v_i} \in V$. While the coordinates of the molecular localizations are a natural choice for node features, using them directly can limit the model's generalization capability due to their absolute positioning. To address this issue, the node features are designed to impose an inductive bias of invariance to the molecules' absolute spatial information by using Laplacian eigenvectors.

Laplacian eigenvectors provide a natural generalization of transformer positional encodings for graphs, equipping each node with a perception of its structural role within the graph[32]. We compute these eigenvectors from the factorization of the graph Laplacian matrix, $\boldsymbol{\Delta}$, defined as:

$$\boldsymbol{\Delta} = \mathbf{I} - \mathbf{D}^{-\frac{1}{2}}\mathbf{A}\mathbf{D}^{-\frac{1}{2}} = \mathbf{U}^T \boldsymbol{\Lambda} \mathbf{U}, \tag{1}$$

where $\mathbf{I}$ is the identity matrix, $\mathbf{A}$ is the $N^v \times N^v$ adjacency matrix (with $N^v$ representing the number of nodes in the graph), $\mathbf{D}$ is the degree matrix, and $\boldsymbol{\Lambda}$ and $\mathbf{U}$ denote the eigenvalues and eigenvectors, respectively. We used the $n = 5$ smallest non-trivial eigenvectors as node features for all experiments. Additionally, we take their absolute values to address the sign ambiguity inherent in eigenvectors. While this choice has been reported to reduce the expressiveness of graph Laplacian eigenvectors in certain cases[49], we did not observe a significant impact on MIRO's performance.

Edge features $\mathbf{e}_{ij} \in E$ encode relational attributes between nodes $i$ and $j$, such as the Euclidean distance and positional displacement describing their relative arrangement. In the current implementation, each edge feature includes the vector $\mathbf{d}_{ij} = \mathbf{x}_j - \mathbf{x}_i$, where $\mathbf{x}_i$ and $\mathbf{x}_j$ are the coordinates of nodes $i$ and $j$, respectively. Although the input graph is undirected—meaning that information flows symmetrically between connected nodes—directional information is preserved by assigning displacement vectors $\mathbf{d}_{ij}$ and $\mathbf{d}_{ji} = -\mathbf{d}_{ij}$ to each message-passing direction. This formulation enables MIRO to capture relative orientations of neighboring nodes while maintaining the symmetry of the graph structure.

This selection of node and edge features allows MIRO to inherently analyze graphs of varying sizes and spatial extents without requiring additional processing complexity. Moreover, the architecture is agnostic to the specific type or number of descriptors used: while distance and directional cues serve as the primary relational information in our current implementation, the framework can readily incorporate additional edge attributes—such as temporal proximity, semantic labels, topological relations, or domain-specific measurements—depending on the needs of the application.

### MIRO architecture

MIRO transforms the input node and edge features into higher-level latent representations $\mathcal{G}$ using a learnable dense layer followed by ReLU activation, mapping $\mathbf{v}_i$ and $\mathbf{e}_{ij}$ into latent vectors $\mathbf{v}'_i$ and $\mathbf{e}'_{ij}$, each with a dimensionality of 256. This latent representation serves as the input to a sequence of $K$ recurrent steps that recurrently update a hidden graph $\mathcal{G}_h^k$. At each recurrent step, the updated hidden nodes features are also decoded through a learnable dense layer and used to calculate the loss function (Fig. 1c).

The core operations are executed within the MIRO block $\mathcal{M}$ (Fig. 1d). At each recurrent step $k$, $\mathcal{M}$ concatenates the latent representations $\mathbf{v}'_i$ and $\mathbf{e}'_{ij}$ with the node $\mathbf{u}_i^k$ and edges $\mathbf{f}_{ij}^k$ of the hidden graph $\mathcal{G}_h^k$, producing $\tilde{\mathbf{u}}_i^k$ and $\tilde{\mathbf{f}}_{ij}^k$ (Fig. 1d). The hidden graph features $\mathbf{u}_i^k$ and $\mathbf{f}_{ij}^k$ are initialized as zeros and, as the recursive process unfolds, they are progressively refined as:

$$\mathbf{f}_{ij}^{k+1} = \phi\left(\left[\tilde{\mathbf{u}}_i^k, \tilde{\mathbf{u}}_j^k, \tilde{\mathbf{f}}_{ij}^k\right]\right), \tag{2}$$

$$\mathbf{u}_i^{k+1} = \psi\left(\sum_{j \in \mathcal{N}_i} \mathbf{f}_{ij}^{k+1}\right), \tag{3}$$

where [,] denotes concatenation, $\mathcal{N}_i$ is the neighborhood of node $i$, and the functions $\phi$ and $\psi$ represent dense layers followed by ReLU activations, which map the output into a 256-dimensional space.

In these operations, the hidden representations play a crucial role in progressively refining the understanding of each node's context within the graph (see Number of recurrent steps: influence on performance and oversmoothing). It is important to note that, for updating the hidden node features $\mathbf{u}_i^{k+1}$, we purely rely on the updated edge hidden states $\mathbf{f}_{ij}^{k+1}$, without including skip connections to the current node hidden states, which is common in various flavors of message passing[30]. The rationale for this choice is to better equip MIRO to discern and emphasize the structural context of each node.

At each recurrent step, the MIRO block further uses a learnable dense layer to decode the updated hidden node features $\mathbf{u}_i^{k+1}$, generating a displacement vector in Cartesian space for each molecular localization. The objective of these learned displacements is, when summed with the localization coordinates, to shift localization belonging to the same cluster toward a common center, resulting in a compact representation of clusters within the SMLM point cloud, while leaving background localizations unaltered.

## MIRO loss function

MIRO is trained on sets of graph representations derived from point clouds reproducing specific molecular organizations. For the clustering task, MIRO is optimized to predict a displacement vector for each molecular localization at each recurrent step. The displacements are learned to shift localizations within the same cluster towards the cluster center, effectively compacting them into well-defined clusters. This problem is formulated as a node regression, with the goal of minimizing the mean absolute error (MAE) between the predicted and ground-truth displacements and inter-localization distances of the displaced positions.

To ensure that the hidden graph representation remains meaningful and to prevent vanishing gradients, at each iteration within an epoch, the loss is further averaged across all recurrent steps. This approach implicitly imposes regularization on the displacement vectors, helping to maintain the structural integrity of the clusters throughout the recurrent step. By calculating the loss at all recurrent steps, MIRO is encouraged to refine the displacement vectors incrementally, preventing early steps from degrading the quality of later predictions and ensuring consistent optimization across the entire sequence of recurrent updates.

For point clouds including only one type of cluster structure, the loss is calculated as the sum of two contributions:

$$\mathcal{L} = \mathcal{L}_{\mathbf{r}} + \mathcal{L}_d. \tag{4}$$

The first term accounts for the difference between ground-truth and predicted displacements and is calculated as:

$$\mathcal{L}_{\mathbf{r}} = \frac{1}{K}\sum_{k=0}^{K-1}\frac{1}{N_v}\sum_{i=0}^{N_v-1}\left|\hat{\mathbf{r}}_i^k - \mathbf{r}_i\right|, \tag{5}$$

where $K$ is the total number of recurrent steps, $N_v$ denotes the number of nodes in the graph, $\hat{\mathbf{r}}_i^k$ is the predicted displacement vector for node $i$ at recurrent step $k$, $\mathbf{r}_i$ is the ground-truth displacement vector for node $i$, and $|\cdot|$ denotes the absolute value.

The second term in Eq. (4) has the objective to minimize the difference between distances of neighbor localizations after adding the target and predicted displacements to the localization coordinates. It is calculated as:

$$\mathcal{L}_d = \frac{\alpha}{K}\sum_{k=0}^{K-1}\frac{1}{N_e}\sum_{(i,j)\in E}\left|d(\hat{\mathbf{p}}_i^k, \hat{\mathbf{p}}_j^k) - d(\mathbf{p}_i, \mathbf{p}_j)\right|, \tag{6}$$

where $\alpha$ is a weighting factor, $E$ represents the set of all pairs $(i, j)$ of neighboring nodes, $d(\cdot, \cdot)$ denotes the Euclidean distance function, and $\hat{\mathbf{p}}^k$ and $\mathbf{p}$ describe the shifted positions after adding the predicted and target displacement to the original localizations. Although both loss functions aim to achieve a similar outcome, we observe that their combined application enhances the model's ability to form compact and well-defined clusters.

Based on this core formulation, additional terms can be introduced depending on the task. For the multiscale clustering depicted in Figs. 1 and 5, the loss is modified by introducing different ground-truth displacements for the steps $[0, k^* - 1]$ and $[k^*, K - 1]$, reflecting clustering at different scales.

In the case of simultaneous clustering and classification of different structures as shown in Fig. 3, the loss function is modified to optimize both spatial clustering and class label. In this scenario, alongside the spatial loss described in Eq. (4), a categorical cross-entropy loss $\mathcal{L}_{CE}$ is added to account for classification performance:

$$\mathcal{L}_{\text{class}} = \frac{\beta}{K}\sum_{k=0}^{K-1}\frac{1}{N_v}\sum_{i=0}^{N_v-1}\mathcal{L}_{CE}\left(\hat{\mathbf{c}}_i^k, \mathbf{c}_i\right), \tag{7}$$

where $\beta$ is a weighting factor, $\hat{\mathbf{c}}_i^k$ is the predicted class label for node $i$ at recurrent step $k$, and $\mathbf{c}_i$ represents the true class label for node $i$.

## MIRO training and augmentations

MIRO supports an effective one- or few-shot learning paradigm, in which training can be performed using as little as a single representative cluster ($N_{c,\text{tr}} = 1$). This is possible due to two key factors: (1) the weak conservation of cluster shape and spatial organization, and (2) a systematic dataset augmentation strategy that promotes generalization across spatial contexts and noise conditions. As demonstrated in the results displayed in Supplementary Table 5, MIRO achieves reasonable performance when trained with just one annotated cluster. An end-to-end example of single-shot training is provided in the GitHub repository https://github.com/DeepTrackAI/MIRO/.

To leverage this data efficiency, MIRO employs an augmentation pipeline that transforms the small set of training clusters ($N_{c,\text{tr}}$) into a large number of diverse point clouds ($N_{\text{pc},\text{tr}}$) for training. Each point cloud is generated by applying a series of transformations to randomly selected clusters, including geometric transformations (rotations, reflections), stochastic perturbations (localization dropout or addition), and spatial jitter (small random displacements). These transformed clusters are then randomly placed within a synthetic FOV to generate the final training samples.

To further mimic realistic imaging conditions, background localizations are added to each point cloud. For non-blinking scenarios, background points are sampled from a uniform spatial distribution. For blinking scenarios, the background is sampled from benchmark data and similarly augmented to reflect variations in localization behavior.

This approach allows the model to learn a robust, transferable representation from minimal annotated input, significantly reducing the need for extensive labeled training data. While the number of clusters and point clouds required depends on shape complexity and density, we found that stable performance could be achieved with small training sets (Supplementary Table 5).

For the benchmark scenarios, training datasets were constructed using all clusters from just three randomly selected point clouds out of the 50 available, with the remaining 47 used for testing. From the clusters contained in these point clouds, we typically generated $N_{\text{pc},\text{tr}} = 1500$ (non-blinking) or $N_{\text{pc},\text{tr}} = 1000$ (blinking) augmented training point clouds per scenario.

## Ablation study and hyperparameter selection

To evaluate the robustness of MIRO and guide its configuration, we performed a series of ablation studies targeting its core hyperparameters and architectural components. These analyses aim to determine the sensitivity of clustering performance to specific design choices and to identify suitable parameter settings across diverse datasets. A summary of the hyperparameters used for each scenario is provided in Supplementary Table 7.

**Number of recurrent steps: influence on performance and oversmoothing.** MIRO adopts a recurrent architecture in which a single graph transformation block is applied multiple times. As previously emphasized, the same set of weights is reused at each step, so increasing the number of recurrent steps does not affect the number of trainable parameters. Instead, this depth controls the effective receptive field of each node, determining how far information can propagate across the graph.

Even in scenarios with a relatively simple structure (e.g., a single clustering level), a sufficient number of recurrent steps may be required to enable long-range interactions.

To demonstrate this effect, we conducted an ablation study on Scenario 6 with blinking, varying the number of recurrent steps from 5 to 25. We quantified clustering quality using the compression index, defined as:

$$\text{Compression Index} = \frac{1}{N_c} \sum_{l=0}^{N_c-1} \left( 1 - \frac{\sum_{i \in \mathcal{C}_l} \| (\mathbf{x}_i + \mathbf{\Delta x}_i) - \boldsymbol{\mu}_l \|_2}{\sum_{i \in \mathcal{C}_l} \| \mathbf{x}_i - \boldsymbol{\mu}_l \|_2} \right) \quad (8)$$

where $N_c$ denotes the number of clusters, $\mathcal{C}_l$ the set of indices for cluster $l$, $\mathbf{x}_i$ the position of point $i$, $\mathbf{\Delta x}_i$ its predicted displacement, and $\boldsymbol{\mu}_l$ the centroid of cluster $l$. A value of compression index close to 1 indicates that the predicted displacements successfully bring the points close to their respective cluster centers (i.e., high compression), whereas a value near 0 reflects little to no compression. This metric quantifies the network's ability to contract clustered localizations toward a common center.

As shown in Supplementary Fig. 2a, the compression index calculated for MIRO consistently increases with increasing depth, flattening around 15 recurrent steps. These results (mean ± standard deviation over 5 runs) indicate that deeper recurrent iterations improve feature integration and enhance clustering compactness.

This behavior also highlights a critical difference between MIRO and conventional message-passing architectures. Classical message-passing GNNs are prone to oversmoothing[43], a phenomenon in which node features become indistinguishable as the number of layers increases. This issue is commonly diagnosed using the Dirichlet energy:

$$\text{Dirichlet Energy} = \frac{1}{N_v} \sum_{i=0}^{N_v-1} \sum_{j \in \mathcal{N}_i} \left\| \mathbf{u}_i - \mathbf{u}_j \right\|_2^2, \quad (9)$$

where $\mathbf{u}_i$ denotes the feature vector of node $i$ at the final recurrent step $K-1$, and $N_v$ is the total number of nodes in the graph[50].

Supplementary Fig. 2a, b compares MIRO to standard message-passing networks of similar depth on the same dataset. While MIRO achieves consistent compression index values and maintains high Dirichlet energy beyond 15 recurrent steps, traditional GNNs exhibit rapid oversmoothing, with both metrics deteriorating as the network deepens. This underscores MIRO's ability to preserve feature diversity and spatial coherence, even with deep processing.

**Dimensionality of the hidden representation.** To assess the appropriate model capacity for MIRO, we performed an ablation study on the dimensionality of the hidden node and edge features.

Our objective was to identify the smallest hidden dimension that consistently guarantees high clustering performance across all tested scenarios. To this end, we selected a challenging synthetic dataset containing mixed shapes (spots and ellipses), where the network must simultaneously learn localization patterns and distinguish between structural classes.

We evaluated dimensionalities of 64, 128, 256, and 512 to identify an optimal hidden size for MIRO's node and edge representations. As shown in Supplementary Table 8, a dimensionality of 256 yields the best overall performance across a broad set of metrics.

Smaller hidden dimensions, such as 64 and 128, result in a minor drop in both clustering quality and spatial resolution. However, performance remains respectable even at 64 dimensions, as the essential structural features are preserved and metric scores remain competitive.

Conversely, increasing the dimensionality to 512 does not lead to significant improvements. In fact, the gains plateau or slightly regress for most metrics, while computational cost and memory usage increase.

These results support the use of a 256-dimensional hidden representation as a robust default configuration for diverse tasks, offering a favorable balance between expressive power and efficiency. They also reinforce the idea that MIRO achieves its performance not through brute-force parameter scaling, but through its effective recurrent architecture and well-structured loss function.

**Loss function weights.** MIRO's loss function consists of multiple components, depending on the task. In all cases, the core objective includes two terms: a displacement loss $\mathcal{L}_\mathbf{r}$, which encourages nodes to move toward their cluster centers, and a distance-preserving auxiliary loss $\mathcal{L}_d$, which maintains relative distances between neighboring points after displacement. The contribution of the latter is modulated by the hyperparameter $\alpha$ (Eq. (4)).

In classification scenarios, such as those involving multiple structural types, a third term is added: a categorical cross-entropy loss $\mathcal{L}_{CE}$, weighted by the hyperparameter $\beta$, which guides the model in predicting shape labels for individual nodes.

To evaluate the sensitivity of performance to these hyperparameters, we conducted ablation studies on a representative dataset. Specifically, we selected a multishape dataset combining circular and elliptical clusters, as it provides a comprehensive test case with diverse structural features.

The results in Supplementary Table 9 demonstrate that MIRO maintains consistently high performance across a broad range of hyperparameter values. For the clustering loss weight $\alpha$, values between 0 and 20 yield comparable results in terms of overall clustering quality, with intermediate settings providing marginal gains in compactness and separation metrics such as ARI and IoU.

At first glance, this robustness might suggest that the auxiliary loss weighted by $\alpha$ is unnecessary. However, its role becomes critical in challenging conditions. As shown in Supplementary Fig. 3, when applied to datasets affected by blinking artifacts, setting $\alpha > 0$ is essential to achieve a stable compression index. This highlights the utility of the auxiliary term in improving robustness under noisy conditions, while also confirming that MIRO's core architecture remains effective even in its absence.

Regarding the classification loss weight $\beta$, performance remains stable for values between 0.05 and 0.5, with F1 scores consistently above 0.92, indicating reliable structural classification. Moderate values of $\beta$ (e.g., 0.15–0.2) yield slight improvements in ARI and $\text{RMSE}_{x,y}$, suggesting that a moderate emphasis on classification enhances spatial precision without compromising clustering performance.

Overall, these findings confirm that MIRO is not highly sensitive to the exact choice of $\alpha$ and $\beta$, simplifying its configuration and highlighting its robustness across multiple simultaneous tasks.

**Training efficiency and practical considerations.** Training time is not substantially affected by the number of recurrent steps, as the model's capacity remains constant, with a total of 528,386 trainable parameters. On an NVIDIA A100 GPU with 40 GB of available memory, training a MIRO model takes approximately 20–60 min per dataset, with variation primarily driven by dataset size and graph density. Memory usage remains modest, owing to the model's shallow architecture and relatively small parameter count.

## Metrics for performance evaluation

Clustering in SMLM presents several challenges that impact the precise quantification of molecular organization and, consequently, the biological insights derived from these data. The effectiveness of a clustering algorithm should be evaluated based on its ability to accurately quantify key parameters, including the number of clusters, their positions, and the number of objects within each cluster.

In our study, we report results obtained using several metrics, as summarized in Table 1, which extend beyond those used in the benchmark study[23]. In particular, we introduce cluster-level metrics, such as the Jaccard Index for cluster detection ($JI_c$), the root mean squared relative error in the number of localizations per cluster ($RMSRE_N$), and the root mean squared error in cluster centroid position ($RMSE_{x,y}$). For their calculation, we first perform a distance-based pairing between clusters of the ground truth and prediction partitions using a Hungarian algorithm[51]. A predicted cluster is considered a true positive (TP) if its centroid is within a threshold distance $\xi$ from that of a ground truth cluster. It is considered a false positive (FP) if it has no corresponding cluster in the ground truth within $\epsilon$. Similarly, a ground truth cluster with no corresponding predicted cluster within $\xi$ is accounted for as a false negative (FN).

We calculate the $JI_c$ as:

$$JI_c = \frac{TP}{TP + FP + FN}. \tag{10}$$

$RMSRE_N$ and $RMSE_{x,y}$ are calculated only for the $N_{TP}$ paired clusters as:

$$RMSRE_N = \sqrt{\frac{1}{N_{TP}} \sum_{i \in TP} \left(\frac{\hat{N}_i - N_i}{N_i}\right)^2}, \tag{11}$$

where $N_i$ and $\hat{N}_i$ represent the number of localizations associated with the ground-truth and predicted $i$-th cluster, and

$$RMSE_{x,y} = \sqrt{\frac{1}{N_{TP}} \sum_{i \in TP} \left((\hat{x}_i - x_i)^2 + (\hat{y}_i - y_i)^2\right)}, \tag{12}$$

where $x_i$, $y_i$, $\hat{x}_i$, and $\hat{y}_i$ represent the coordinates of the ground-truth and predicted $i$-th cluster, respectively. These metrics assess the accuracy of molecule assignment within clusters and provide insights into the clustering method's ability to correctly determine the number of clusters, their size, and their position.

Moreover, particularly in cases with a significant imbalance between cluster sizes, metrics like ARI can fail to reflect the true performance of an algorithm. In fact, although ARI is a widely used metric for assessing the agreement between two partitions[36] and is considered a standard tool in cluster validation, it is sensitive to cluster size imbalances. As discussed in the literature[37,38,52,53], ARI tends to emphasize agreement on larger clusters while providing limited insights into the agreement for smaller clusters. This imbalance issue is particularly relevant in SMLM data, where a substantial number of non-clustered molecules are often treated as a "background" cluster[23]. In scenarios described in ref. 23, the ratio between non-clustered and clustered localizations is at most 1:1 (being in some cases 4:1), meaning that the number of molecules with spatial organization is at most equal

to those contributing to the background. For example, in Scenario 6, with 1000 non-clustered localizations and 1000 clustered localizations divided into 20 clusters of 50 each, the "background" cluster contains 20 times more localizations than the individual clusters. Metrics like ARI in such cases primarily reflect the agreement for the larger clusters and offer limited information about the smaller clusters.

To mitigate the effects of cluster size imbalance, Romano et al.[37] suggested using the AMI, leaving the ARI for balanced cases. Warrens and van der Hoef[38] proposed a variant of the adjusted Rand Index, ARI[†] to provide a more robust measure of clustering performance. Recently, Saavedra et al.[22] used a modified ARI that excludes non-clustered molecules from the calculation and focuses solely on evaluating the similarity of the partitions constituting the actual clusters ($ARI_c$).

For the evaluations conducted in this article, we utilized all the aforementioned metrics, and MIRO consistently demonstrates enhanced performance (Table 1).

**Comparison of ARI-like metrics.** To demonstrate the practical advantages of using AMI, ARI[†], and $ARI_c$ over ARI, we provide a toy example based on the data of Scenario 6. Let us consider two clustering methods, A and B. Method A identifies the correct number of clusters (20) but misassigns 20% of background localizations to the clusters while missing 10% of clustered localizations and assigning them to the background. The corresponding confusion matrix is shown in Table 2.

Method A accurately identifies the number of clusters, yielding $JI_c$ of 1, and slightly overestimates the number of localizations per cluster by 10% ($RMSRE_N = 0.1$). Summary scores provide ARI = 0.62, ARI[†] = 0.67, AMI = 0.7, and $ARI_c = 0.8$.

Conversely, Method B has a larger error in assigning clustered localizations to the background (18%) and breaks down half of the ground truth clusters into two, resulting in a 33% of false positive clusters ($JI_c = 0.67$) and a significant underestimation of the number of localizations per cluster ($RMSRE_N = 0.34$). As such, it provides an inaccurate view of the cluster organization. However, it better recognizes the background localizations, with a 10% error. The corresponding confusion matrix is shown in Table 3. Although Method B performs worse in cluster quantification, due to the imbalance, it achieves a higher ARI = 0.65. In contrast, the other metrics better reflect its actual performance, providing smaller values as compared to method A (ARI[†] = 0.47, AMI = 0.67, and $ARI_c = 0.46$).

In scenarios with significant imbalances, such as those encountered in SMLM, ARI[†], AMI, and $ARI_c$ provide a more accurate assessment of clustering performance compared to traditional ARI. ARI is largely affected by the classification between clustered and non-clustered localization and does not reflect the accuracy in determining the actual organization of small clusters.

These metrics also have some limitations. For example, by excluding the background, $ARI_c$ does not account for the false positive assignment of non-clustered localizations to clusters. It is also important to note that in specific cases, ARI[†] can be highly sensitive to the misassignment of cluster localizations, resulting in very low scores even for minor errors. This behavior, along with the sensitivity of ARI to cluster size imbalance, stems from the calculation methods of ARI and ARI[†]. The ARI is the harmonic mean of the adjusted Wallace indices, which are weighted means of cluster indices. These weights are quadratic functions of cluster sizes, which leads to increased susceptibility to size imbalances. Conversely, ARI[†] uses ordinary averages instead of weighted means. While this approach reduces the impact of misassigning non-clustered localization, it increases the sensitivity to small errors in the misassignment of clustered localizations, especially when the number of clusters and clustered localizations is very low compared to the background.

**Table 2 | Confusion matrix for method A**

|  | Non-clust | Cl 1 | Cl 2 | Cl 3 | Cl 4 | … | Cl 20 |
|---|---|---|---|---|---|---|---|
| Non-clust (GT) | 800 | 10 | 10 | 10 | 10 | … | 10 |
| Cl 1 | 5 | 45 | 0 | 0 | 0 | … | 0 |
| Cl 2 | 5 | 0 | 45 | 0 | 0 | … | 0 |
| Cl 3 | 5 | 0 | 0 | 45 | 0 | … | 0 |
| Cl 4 | 5 | 0 | 0 | 0 | 45 | … | 0 |
| ⋮ | ⋮ | ⋮ | ⋮ | ⋮ | ⋮ | ⋱ | ⋮ |
| Cl 20 | 5 | 0 | 0 | 0 | 0 | … | 45 |

**Table 3 | Confusion matrix for method B**

|  | Non-clust | Cl 1 | Cl 2 | … | Cl 11 | Cl 12 | … | Cl 29 | Cl 30 |
|---|---|---|---|---|---|---|---|---|---|
| Non-clust (GT) | 900 | 5 | 5 | … | 5 | 0 | … | 5 | 0 |
| Cl 1 | 9 | 41 | 0 | … | 0 | 0 | … | 0 | 0 |
| Cl 2 | 9 | 0 | 41 | … | 0 | 0 | … | 0 | 0 |
| ⋮ | ⋮ | ⋮ | ⋮ | ⋱ | ⋮ | ⋮ | ⋱ | ⋮ | ⋮ |
| Cl 11 | 9 | 0 | 0 | … | 21 | 20 | … | 0 | 0 |
| ⋮ | ⋮ | ⋮ | ⋮ | ⋱ | ⋮ | ⋮ | ⋱ | ⋮ | ⋮ |
| Cl 20 | 9 | 0 | 0 | … | 0 | 0 | … | 21 | 20 |

To mitigate these issues, we decided to comparatively report multiple metrics. Nevertheless, a potential improvement could involve developing new ARI-like metrics that employ weighted averages with weights that are less sensitive to cluster size.

## DBSCAN parameter selection

In the benchmark study[23], distinct DBSCAN parameter sets ($\varepsilon$ and minPts) were used to maximize ARI or IoU, independently. While this approach may be appropriate for assessing theoretical performance limits, it raises important practical concerns. Most notably, it becomes unclear which set of parameters should be adopted in real-world applications, as the optimal configuration for ARI may differ significantly from that for IoU — an issue clearly illustrated in the scenarios with blinking, where the two metrics are optimized by substantially different parameter choices (see Supplementary Table 1 in ref. 23).

Furthermore, although the benchmark reports only a single parameter pair per scenario and metric, the best metric values reported in their comparison appear to have been derived from an optimization procedure conducted over each FOV. Specifically, the authors refer to results in Figs. 3 and 4 as the "average peak score" or "mean of the maximal ARI and IoU scores", suggesting that the best metric values were obtained by tuning parameters independently for each FOV before averaging the results. This interpretation is supported by our own attempts to reproduce the reported values, which provided comparable results only when optimizing parameters at the level of individual FOVs, yielding a range of $\varepsilon$ and minPts values across each scenario.

While such FOV-specific optimization may be informative for gauging the best-case performance of a method, it is not feasible in real experimental settings, where ground truth is unavailable. Moreover, it provides little guidance to practitioners on how to select suitable parameters for a new dataset.

To ensure a realistic and fair comparison, we adopted a consistent strategy: for each scenario, a single pair of DBSCAN parameters was applied uniformly across all FOVs. The selected parameters were those that maximized the ARI metric. For the benchmark datasets analyzed with DBSCAN, we used the DBSCAN settings that achieved the highest ARI scores as reported in Supplementary Table 1 in ref. 23.

## Simulated data

MIRO was validated on all the datasets (both without and with blinking) described in ref. 23, with clustered localizations generated in $2000 \times 2000$ nm² regions. For the results described in Fig. 2 and Table 1, we used:

Scenario 5: 100 clusters of 15 molecules per cluster with 50% of molecules being clustered.

Scenario 6: 20 elliptically shaped clusters with aspect ratio 3:1 and each having 50 molecules, with 50% of the total molecules being clustered.

Scenario 8: 10 clusters with 5 molecules per cluster and 10 clusters with 15 molecules per cluster, with 50% of the total molecules clustered.

For datasets with blinking, for each molecule in the original dataset 4–5 localizations on average were generated and distributed according to a 2-dimensional normal distribution centered at the molecule position, with a standard deviation corresponding to the localization precision.

In Supplementary Tables 3 and 4, we report the results obtained for all scenarios, without and with blinking, respectively. The additional scenarios are:

Scenario 2: 20 clusters of 15 molecules per cluster with 50% of the total molecules being clustered.

Scenario 3: 20 clusters of 15 molecules per cluster with 20% of the total molecules being clustered.

Scenario 4: 20 clusters of 5 molecules per cluster with 25% of the total molecules being clustered.

Scenario 7: 10 clusters with a width of 25 nm and 10 clusters with a width of 75 nm, with 50% of the total molecules clustered.

Scenario 9: 10 clusters with 15 molecules per cluster and a cluster width of 25 nm, and a further 10 clusters with 135 molecules and a cluster width of 75 nm, thus maintaining molecule density with increased size, with 50% of the total molecules clustered.

Scenario 10: 20 clusters of 15 molecules per cluster with 50% of the total molecules being clustered, but with non-clustered molecule density increasing from left to right across the region.

We simulated two further scenarios:

C-shaped clusters: $6400 \times 6400$ nm² images, with 30–60 clusters per image obtained by randomly placing localizations on semicircles with a radius of 250 nm and a radial standard deviation of 50 nm. Each cluster had a random number of localizations between 30 and 60, drawn from a uniform distribution. The number of non-clustered molecules was 6% of the total number of localizations, corresponding to 73% of the total number of structures (i.e., the sum of clusters and non-clustered localizations).

Ring-shaped clusters: $6400 \times 6400$ nm² images, with 60–70 clusters per image obtained by randomly placing localizations on circles with a radius of 250 nm and a radial standard deviation of 50 nm. Each cluster had a random number of localizations between 60 and 80, drawn from a uniform distribution. The number of non-clustered localizations was 7% of the total number of localizations, corresponding to 84% of the total number of structures.

In addition, for the evaluation of MIRO's performance in resolving adjacent clusters, we simulated groups of localizations distributed according to a 2-dimensional normal distribution with a standard deviation of 25 nm. The number of localizations per cluster was drawn from a geometric distribution with an average of 90.

For the training of MIRO's model used for the analysis of integrin organization, we simulated groups of localizations distributed according to a 2-dimensional normal distribution with a standard deviation of 25–40 nm within $10,000 \times 10,000$ nm² regions. The number of localizations per cluster was drawn from a geometric distribution with an average of 25. The number of non-clustered localizations was 4% of the total number of localizations, corresponding to 60% of the total number of structures.

For the proof of principle of the multiscale clustering, symmetric clusters having a random number of localizations between 6 and 30, drawn from a uniform distribution, with positions drawn from a 2-dimensional normal distribution with a standard deviation of 15 nm, were arranged with a 5 to 9-fold symmetry along a circle of radius 40 nm to form 40–60 rings. The number of non-clustered localizations was 1.5% of the total number of localizations, corresponding to 63% of the total number of structures. Images were $1250 \times 1250$ nm$^2$.

For the training of MIRO for the analysis of the NPC, we generated $1250 \times 1250$ nm$^2$ images containing 5–9 NPC-like structures. Each structure was composed of 8 corners with a common vertex. For distributing the localizations, each corner was approximated as an isosceles triangle with height $h$, which defined the circle radius as $r = h/2 = 50$ nm. A random number of localizations from a uniform distribution between 0 and 80 were placed according to a normal distribution centered in the triangle centroid, with an angle-dependent standard deviation given by the center-to-perimeter distance divided by 1.8, providing an effective inner and outer radius of 20 and 80 nm, respectively. The number of non-clustered localizations was 3% of the total number of localizations, corresponding to 89% of the total number of structures.

## Experimental data

The experimental dataset used for the analysis of $\alpha5\beta1$ integrin organization was obtained by dSTORM imaging of a HeLa cell line modified to express ITGA5-HaloTag. In brief, cells were genetically edited with CRISPR/Cas9 to insert the HaloTag coding sequence at the 3′ terminus of ITGA5 cDNA. HeLa cells were co-transfected with a Cas9 and guide RNA-containing plasmid (lentiCRISPRv2 vector) and a repair plasmid (pBluescript) carrying the ITGA5-HaloTag cDNA, using polyethylenimine as the transfection reagent. Two days post-transfection, cells were treated with 1 μg/ml puromycin for 3 days to select transfected cells, as lentiCRISPRv2 includes a puromycin resistance gene. Surviving cells were subsequently expanded for 2 weeks and subjected to three rounds of cell sorting.

Cells ($2 \times 10^4$) were seeded on fibronectin-coated (10 μg/ml) glass-bottom dishes and allowed to adhere overnight. For staining, cells were incubated in a growth medium containing 200 nM Janelia Fluor® 647 HaloTag® Ligand for 30 min at 37 °C, followed by three washes with fresh growth medium. An additional 1-h incubation allowed for the removal of unbound ligand, followed by another three washes. Cells were then fixed in 2% paraformaldehyde for 20 min at room temperature.

Single-molecule fluorescence imaging was carried out in a STORM imaging buffer with an inverted microscope (DMi8, Leica) equipped with a TIRF-illumination module (Infinity TIRF High Power, Leica) and a CMOS camera (Photometrics 95B). The beam of a 638-nm laser diode (180 mW, LBX-638, Oxxius) was combined with the beam of a 405-nm laser diode (50 mW, LBX-405, Oxxius) and coupled into the microscope through a polarization-maintaining monomode fiber. A $100 \times$ objective with a numerical aperture of 1.47 (HC PL APO 100 × /1.47, Leica) was used for TIRF illumination. The excitation beam was reflected into the objective by a quad-line dichroic beamsplitter and the fluorescence was detected through a quadruple band pass filter (set TRF89902 ET, Chroma). Photoactivation was manually controlled by the output power of the 405 nm laser and applied in adequate pulses. Fluorescence imaging was performed by excitation at 638 nm (0.2–0.5 kW/cm$^2$). The camera was operated at a frame rate of 27 Hz. Experiments were performed in three independent biological replicates.

Images were processed with ThunderSTORM[54]. Data were fitted with a symmetric Gaussian PSF model using maximum likelihood estimation. The $x$ and $y$ localization positions were corrected for residual drift by an algorithm based on cross-correlation. Localizations were filtered by the localization precision (<30 nm) and to exclude dim and very bright localizations (120 < counts < 750). Localizations

persistent over consecutive frames detected within 40 nm from one another were merged into one localization.

Experimental data for the NPC were obtained from ref. 13.

## Comparison with a supervised graph-based clustering framework

To compare MIRO with recent supervised graph-based clustering methods, we implemented a two-stage GNN pipeline inspired by the approach described in ref. 22, hereafter referred to as MAGIK-S. Clustering performance was evaluated on two representative benchmark configurations (Scenarios 5 and 6), as reported in Supplementary Table 6.

MAGIK-S consists of two stages. In the first stage, a GNN classifies each localization as clustered or non-clustered, serving as a background filter. In the second stage, a separate GNN processes the filtered point cloud and predicts binary edge labels indicating whether each edge connects localizations belonging to the same cluster. These predictions are used to prune the graph, and final clusters are extracted using the Louvain community detection algorithm. Both GNNs employ the MAGIK architecture[31].

The input graph construction in Stage 1 matches that of MIRO, with the exception that absolute spatial coordinates are used as node features. Stage 2 operates only on localizations predicted as clustered in Stage 1.

To ensure a fair comparison, MAGIK-S was configured to use the same latent dimension (256) as MIRO and three Fingerprinting Graph Neural Network (FGNN) layers, each incorporating four attention heads. The training procedure was the one described in ref. 22, whereas the dataset configuration followed the one used for MIRO.

## Reporting summary

Further information on research design is available in the Nature Portfolio Reporting Summary linked to this article.

## Data availability

The benchmark simulations from ref. 23 used in this study are publicly available at https://github.com/DJ-Nieves/ARI-and-IoU-cluster-analysis-evaluation. The NPC dataset from ref. 13 is publicly available at https://www.ebi.ac.uk/biostudies/BioImages/studies/S-BIAD8. Source Data are provided with this paper.

## Code availability

The code used to develop the model, perform the analyses, and generate the results of this study is publicly available and has been deposited in GitHub at https://github.com/DeepTrackAI/MIRO/, under MIT license. The specific version of the code associated with this publication is archived in Zenodo and is accessible at https://doi.org/10.5281/zenodo.17194652[55].

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

## Acknowledgements

J.P. acknowledges funding from the Adlerbertska Research Foundation (Grant AF2024-0305). G.V. acknowledges support from the Horizon Europe ERC Consolidator Grant MAPEI (grant number 101001267) and the Knut and Alice Wallenberg Foundation (grant number 2019.0079). C.M. acknowledges support through grant RYC-2015-17896 funded by MCIN/AEI/10.13039/501100011033 and "ESF Investing in your future", grants BFU2017-85693-R and PID2021-125386NB-I00 funded by MCIN/AEI/10.13039/501100011033/ and "ERDF A way of making Europe".

## Author contributions

J.P., G.V., and C.M. conceived the study. J.P. and C.M. conducted the study and performed formal analysis. J.P. designed and developed the computer code. S.M.-O., M.M., and J.B. performed experiments. J.B., M.G., G.V., and C.M. supervised the study. J.P., G.V., and C.M. wrote the manuscript with input from all authors.

## Funding

## Competing interests

J.P., M.G., G.V., and C.M. hold shares and/or stock options of the company IFLAI AB. The other authors declare no competing interests.
