## [Transparent Peer Review file · Nature Communications]

Enhanced Spatial Clustering of Single-Molecule Localizations with Graph Neural Networks

Corresponding Author: Professor Giovanni Volpe

Version 0:

Reviewer comments:

Reviewer #1

(Remarks to the Author)

The authors introduce a representation clustering method, MIRO, based on an rGNN architecture with one/few-shot learning. MIRO transforms the original spatial graph, constructed using Delaunay triangulation for edges and Laplacian positional encoding for node features, into a representation that is more amenable to standard clustering algorithms such as DBSCAN. This is achieved by iteratively shifting nodes belonging to the same cluster toward a common center at each recurrent step. This transformation facilitates parameter tuning in DBSCAN, enabling more consistent clustering across datasets with varying cluster shapes, densities, and localization counts. Benchmarking on various synthetic and real datasets demonstrates the high performance of MIRO as a representation learning step preceding clustering. The paper is generally well-written, with only a few vague sentences and references requiring clarification (see below). However, some minor improvements could further enhance the manuscript.

Major Comments:

1. Certain aspects of MIRO's architecture and learning process require further clarification:
 - How is the number of recurrent layers in MIRO determined? I assume it relates to the hierarchical clustering structure. If only a single hierarchical step exists (i.e., nodes and their respective clusters), would only one layer be required?
 - While node features in the hidden layers are not directly aggregated from neighboring nodes but instead updated via edges, the edge features themselves are updated based on their two endpoints. This suggests that node features still incorporate information from their neighbors indirectly. Would this still be considered message passing? If so, could MIRO be susceptible to oversmoothing? An ablation study investigating MIRO's performance with varying hidden layer depths could help address this concern.
 - How was the dimensionality of 256 chosen for the representation space?
 - Is MIRO trained in a supervised manner (i.e., using ground truth clustering labels) to predict the displacement vector?
2. In the section on multiscale clustering of the nuclear pore complex (and Fig. 5c-f), it would be beneficial to include a comparison between MIRO and DBSCAN using a real dataset, similar to the comparison provided for synthetic data.
3. While the authors justify the comparison between MIRO and DBSCAN based on DBSCAN's prevalence in the field, including comparisons with other standard clustering techniques would provide additional context for readers.
4. Although MIRO is described as a one/few-shot learning method, the manuscript does not explain the mechanism behind this learning paradigm. Clarifying this would strengthen the contribution.
5. The manuscript mentions multimodal data integration, but this concept is not elaborated upon. Further explanation would be helpful.

Minor Comments:

6. Line 109: The phrase "relational information" appears to refer to physical proximity among localizations. The sentence should explicitly state this for clarity.
7. Line 157: How is the direction vector encoded in the edge features? Does this imply that the input graph is directed? If so, how is the direction of edges determined, and what is its significance?
8. Line 165: It should be explicitly stated that the hidden graph is initialized with node and edge features set to zero.
9. Throughout the manuscript, the authors use the term "linear transformation" to describe data transformations. However, since these transformations are followed by a ReLU activation, they should be referred to as "nonlinear transformations."
10. Line 178: The manuscript should clarify that the loss is calculated at each recurrent step, rather than at each iteration, training step, or epoch, to avoid potential misunderstandings.

11. Line 256: The authors suggest using visual inspection of MIRO's output to guide DBSCAN parameter selection. This process should be described in more detail.

12. Line 356: The manuscript states that cluster shape is encoded as node features. However, since shape is a property of the cluster as a whole rather than individual nodes, further clarification is needed on how this information is incorporated at the node level.

Overall, this is a well-written manuscript with a novel approach to representation clustering. Addressing the points above would further strengthen the clarity and impact of the work.

(Remarks on code availability)

Reviewer #2

(Remarks to the Author)

In this paper, the authors present MIRO (Multimodal Integration through Relational Optimization), a novel few-shot (or one-shot) geometric deep learning framework leveraging recurrent graph neural networks (rGNNs) to transform point cloud data for more efficient clustering. By learning to “squeeze” localized points around a common center, MIRO preserves the overall structural relationships within a dataset while enhancing the separation between clustered and non-clustered points, thereby simplifying parameter selection for conventional clustering methods like DBSCAN and supporting structures of varying scales simultaneously. This approach proves effective across a range of single-molecule localization microscopy (SMLM) datasets.

However, the authors need to clarify and support the methodological details and underlying rationales—particularly regarding how MIRO maintains robust performance across diverse scales and complex noise patterns—with more concrete evidence.

1. The authors present MIRO primarily as a booster for existing clustering algorithms, yet their evaluation compares only scenarios with and without MIRO in conjunction with DBSCAN. To strengthen the manuscript, it would be fair to benchmark MIRO's performance against other clustering “boosters” or enhanced variations of DBSCAN, such as HDBSCAN or DBSCAN++, ensuring fair comparison. Going through the literature, some methods tackle similar problems as MIRO (e.g. <https://www.nature.com/articles/s41467-024-46106-0>).

2. Although DBSCAN requires two hyperparameters, the MIRO algorithm itself has at least two (and in certain scenarios, three) hyperparameters for its loss function, ground-truth displacement parameter k , and the cross-entropy weighting factor—resulting in a total of three to five parameters to tune. The authors should provide additional details on MIRO's training process, including assessing its robustness to different hyperparameter values, typical ranges, training time, and practical considerations like memory usage and runtime.

3. A key advantage proposed by the authors is MIRO's capacity for single- or few-shot learning, yet there appears to be no direct evidence in the manuscript demonstrating this. Clarification is needed on whether the reported results were obtained under a single-/few-shot strategy or a more conventional training setup. The authors are encouraged to address how the model's performance scales with reduced training data, discussing any trade-offs in accuracy versus efficiency.

4. While the choice to benchmark against DBSCAN follows a previous comprehensive study (<https://www.nature.com/articles/s41592-022-01750-6>), the authors only validated MIRO on four of the nine simulation datasets from that work. More information on why these four were selected—and why the remaining five were not included—would help clarify the completeness of the results. Ideally, the authors could expand their evaluation to all nine datasets or provide a strong rationale for their selection criteria.

5. Regarding reproducibility, the authors should consider making an end-to-end pipeline available in their GitHub repository rather than only sharing the final clusters used in the evaluation. Providing the raw data, preprocessing steps, training scripts, and final evaluation metrics would enable other researchers to replicate the study's findings precisely and further validate MIRO's effectiveness.

(Remarks on code availability)

The authors have published their code along with a brief tutorial on how to use it. However, the materials provided for replicating the paper's results only include CSV files containing the final clustering indices and data point locations. There is no accompanying code or pipeline that demonstrates how to preprocess the data, train the model, and replicate the paper's findings end to end.

Version 1:

Reviewer comments:

Reviewer #1

(Remarks to the Author)

The authors have thoroughly addressed all of my previous concerns. I have no further comments and appreciate the improvements made to the manuscript.

(Remarks on code availability)

Reviewer #2

(Remarks to the Author)

I appreciate the authors' response, and here are my comments:

1. Since the MIRO algorithm is trained in a supervised manner with ground-truth labels, even if it serves only as a pre-processing step for a clustering algorithm, it effectively makes MIRO a supervised clustering approach. Therefore, the result should be improved to some extent. However, the SMLM task, as I understand it, is intended to investigate potential clusters without prior labels. In this case, please clarify the rationale for making the approach supervised. In the benchmark study the authors refer to, only 1 out of 7 methods is supervised. If the authors wish to demonstrate the strength of MIRO, they might consider comparing it against some advanced supervised clustering algorithms.

2. The authors have provided an appropriate explanation to my earlier question. One minor suggestion: perhaps the GitHub repository's Jupyter notebook could be provided as a fully executed version, so that users can see the expected results directly.

(Remarks on code availability)

Version 2:

Reviewer comments:

Reviewer #2

(Remarks to the Author)

The authors have clearly addressed my previous concerns. I have no further comments.

(Remarks on code availability)

Point-by-point response to Reviewers of *Enhanced Spatial Clustering of Single-Molecule Localizations with Graph Neural Networks*

Pineda et al.

Response to Reviewer 1

Comment 1

The authors introduce a representation clustering method, MIRO, based on an rGNN architecture with one/few-shot learning. MIRO transforms the original spatial graph, constructed using Delaunay triangulation for edges and Laplacian positional encoding for node features, into a representation that is more amenable to standard clustering algorithms such as DBSCAN. This is achieved by iteratively shifting nodes belonging to the same cluster toward a common center at each recurrent step. This transformation facilitates parameter tuning in DBSCAN, enabling more consistent clustering across datasets with varying cluster shapes, densities, and localization counts. Benchmarking on various synthetic and real datasets demonstrates the high performance of MIRO as a representation learning step preceding clustering. The paper is generally well-written, with only a few vague sentences and references requiring clarification (see below). However, some minor improvements could further enhance the manuscript.

Reply to Comment 1

We thank the Reviewer for their positive and constructive feedback. We are glad that the Reviewer found the manuscript well-written and appreciate the suggestions for improving its clarity, which we have addressed below.

Major Comments:

Comment 2

1. Certain aspects of MIRO's architecture and learning process require further clarification:
 - How is the number of recurrent layers in MIRO determined? I assume it relates to the hierarchical clustering structure. If only a single hierarchical step exists (i.e., nodes and their respective clusters), would only one layer be required?

Reply to Comment 2

We thank the Reviewer for pointing out the need to clarify this aspect. In MIRO, the architecture consists of a single layer (the MIRO block) that is applied recurrently. To avoid confusion, we talk about recurrent steps instead of recurrent layers. The number of recurrent steps does not affect the model capacity in terms of learnable parameters. Instead, the number of recurrent steps defines the size of the receptive field and therefore needs to be adapted to the spatial scale and complexity of the

clustering problem.

We now clarify this point in the revised manuscript, where we state:

We emphasize that MIRO uses a single-layer architecture. As a result, increasing the number of recurrent steps does not affect the number of learnable parameters. Instead, the number of recurrent steps defines the size of the receptive field and therefore needs to be adapted to the density of the point cloud and the complexity of the clustering problem, as discussed in “Number of recurrent steps: influence on performance and oversmoothing”.

Even in cases with a relatively simple structure (e.g., a single level of clustering), multiple recurrent steps may be necessary to propagate information effectively across the graph. To support this point, we have added a new section in the Methods entitled “Ablation study and hyperparameter selection”. In the subsection “Number of recurrent steps: influence on performance and oversmoothing”, we present the results of an ablation study in which we vary the number of recurrent steps and evaluate clustering performance on data from Scenario 6 with blinking. In Supplementary Figure 2, we report both the *compression index*, which quantifies the extent to which node features are drawn toward common cluster centers, and the *Dirichlet energy*, which reflects the sharpness of feature transitions between neighboring nodes. The results show that increasing the number of recurrent steps improves both cluster compactness and feature discriminability. We also provide a direct comparison with standard message-passing networks, with and without layer normalization, as further discussed in the reply to the next comment.

Comment 3

- While node features in the hidden layers are not directly aggregated from neighboring nodes but instead updated via edges, the edge features themselves are updated based on their two endpoints. This suggests that node features still incorporate information from their neighbors indirectly. Would this still be considered message passing? If so, could MIRO be susceptible to oversmoothing? An ablation study investigating MIRO’s performance with varying hidden layer depths could help address this concern.

Reply to Comment 3

We thank the Reviewer for this observation. Battaglia et al. (arXiv:1806.01261, 2018) formalize a general framework for graph neural networks that defines message passing as the process of computing messages along the edges of a graph and aggregating them at nodes. This formulation is broader than the graph convolutional networks earlier described by Kipf & Welling (arXiv:1609.02907, 2016; ICLR 2017), which do not explicitly model edge features. In this broader sense, MIRO can indeed be considered a form of message passing.

However, MIRO’s architecture differs from conventional message-passing networks in key ways. Notably, it avoids stacking multiple layers that aggregate neighbor features at each depth. Instead, MIRO applies a single transformation recurrently. Moreover, each recurrent step processes both the original graph

structure and the evolving hidden representation, preserving initial semantic information. These design choices are central to preventing oversmoothing.

To assess the risk of oversmoothing, we performed an ablation study evaluating both oversmoothing and clustering performance as a function of the number of recurrent steps using Scenario 6 with blinking. The discussion of this study is reported in the new section “Number of recurrent steps: influence on performance and oversmoothing”.

In brief, we observed that performance improves with increasing steps and then saturates, with no evidence of degradation — indicating that MIRO maintains node-level feature diversity even after several recurrent steps (Supplementary Figure 2).

This behavior contrasts sharply with standard message-passing networks, both with and without layer normalization. In those models, increasing network depth leads to a rapid decline in both the Dirichlet energy and the compression index, consistent with oversmoothing.

Comment 4

- How was the dimensionality of 256 chosen for the representation space?

Reply to Comment 4

The choice of a 256-dimensional hidden representation was motivated by the need to balance model capacity and computational efficiency. Our goal was to identify the smallest hidden size that consistently ensures high performance across all tested scenarios.

To support this choice, we conducted an ablation study assessing how hidden dimensionality affects clustering performance. For this analysis, we selected a challenging synthetic scenario involving mixed shapes (spots and ellipses), where the network must learn not only localization patterns but also shape classification.

We evaluated hidden dimensions of 64, 128, 256, and 512. While 64-dimensional embeddings already provide reasonable performance, increasing the dimensionality to 256 yields a substantial improvement and robust results across the dataset. Further increasing the dimensionality to 512 brings negligible performance gains while incurring additional computational cost. Based on these findings, we selected 256 as the optimal trade-off between accuracy and efficiency.

The discussion of this study is reported in the new section “Dimensionality of the hidden representation” and full results are presented in Supplementary Table 7.

Comment 5

- Is MIRO trained in a supervised manner (i.e., using ground truth clustering labels) to predict the displacement vector?

Reply to Comment 5

Yes, MIRO is trained in a supervised manner using cluster labels, which are used to derive displacement vectors serving as ground truth for learning.

We have clarified this important point in the revised manuscript. In particular, the Introduction now states:

In this paper, we introduce a novel supervised approach to enhance the versatility of clustering algorithms.

Additionally, we have revised the Methods to explicitly describe the supervised nature of the training process.

Comment 6

2. In the section on multiscale clustering of the nuclear pore complex (and Fig. 5c-f), it would be beneficial to include a comparison between MIRO and DBSCAN using a real dataset, similar to the comparison provided for synthetic data.

Reply to Comment 6

We have added a new figure (Supplementary Figure 1) presenting a direct comparison between MIRO and DBSCAN on the real nuclear pore complex dataset. The results confirm that it is challenging to identify optimal DBSCAN parameters in this setting. In particular, DBSCAN often leads to cluster fragmentation, which is reflected in a peak of small-sized clusters in the resulting distribution. In contrast, MIRO provides more consistent and coherent clustering results. We have updated the manuscript text accordingly. It now states:

These results highlight the accuracy and reliability of MIRO in multiscale real-data applications. In contrast, DBSCAN struggles in this scenario. As shown in Supplementary Figure 1, identifying optimal DBSCAN parameters is non-trivial, and the algorithm often produces fragmented clusters, resulting in an artificial peak of small-sized clusters in the size distribution. Overall, these findings underscore the advantage of MIRO not only over traditional clustering approaches like DBSCAN, but also compared to the task-specific, multi-step analysis pipeline originally proposed for this dataset [13].

Comment 7

3. While the authors justify the comparison between MIRO and DBSCAN based on DBSCAN's prevalence in the field, including comparisons with other standard clustering techniques would provide additional context for readers.

Reply to Comment 7

Our decision to compare MIRO with DBSCAN is not only motivated by its widespread use in the field, but also by findings from a recent comprehensive benchmarking study (Ref. [23] in the revised manuscript), which demonstrated that DBSCAN outperforms other standard clustering algorithms, particularly in the context of single-molecule localization microscopy data. Replicating those findings would offer limited additional value. Moreover, alternative methods not included in that benchmark — such as those suggested by Reviewer 2 — have not demonstrated improved performance over DBSCAN according to recent studies (Refs. [27, 42]).

This rationale is clearly stated in the manuscript:

We selected DBSCAN for this comparison due to its top performance in benchmark studies [23, 27] and its widespread use in the literature [28].

Given the existence of this thorough benchmark and the top results obtained by DBSCAN, we considered it unnecessary to replicate its results by including a comparison with additional clustering methods.

Comment 8

4. Although MIRO is described as a one/few-shot learning method, the manuscript does not explain the mechanism behind this learning paradigm. Clarifying this would strengthen the contribution.

Reply to Comment 8

We have revised the section “MIRO training and augmentations” to clarify the mechanisms that enable MIRO to operate in a one/few-shot learning regime. Specifically, we now better explain the training strategy and augmentations used during representation learning, which allow MIRO to generalize well even when trained with only a single annotated graph.

To support this, we refer the Reviewer to Supplementary Table 5, which presents an ablation study evaluating performance as a function of the number of training clusters for representative scenarios.

Scenario	N	ARI [†]	IoU	JL _c	RMSRE _{N}	RMSE _{x,y}	AMI	ARI _c	ARI
Scenario 5	1	0.560 ± 0.007	0.614 ± 0.005	0.882 ± 0.008	0.54 ± 0.04	3.16 ± 0.05	0.658 ± 0.004	0.62 ± 0.03	0.576 ± 0.011
	300	0.566 ± 0.005	0.612 ± 0.008	0.870 ± 0.012	0.57 ± 0.04	3.16 ± 0.03	0.654 ± 0.005	0.64 ± 0.03	0.560 ± 0.012
Scenario 6	1	0.726 ± 0.009	0.662 ± 0.013	0.94 ± 0.02	0.24 ± 0.02	3.6 ± 0.2	0.788 ± 0.004	0.87 ± 0.02	0.752 ± 0.004
	60	0.724 ± 0.005	0.680 ± 0.010	0.94 ± 0.02	0.21 ± 0.02	3.59 ± 0.11	0.784 ± 0.005	0.864 ± 0.009	0.760 ± 0.007

Supplementary Table 5 Performance metrics comparing single-sample and full-dataset settings for representative scenarios. Each value is reported as mean ± standard deviation over 5 independently trained models.

The results demonstrate that MIRO achieves high clustering performance even in the single-shot case ($N = 1$), with only modest improvements as the number of training graphs increases. We believe this analysis substantiates the one/few-shot learning capabilities of our approach.

Comment 9

5. The manuscript mentions multimodal data integration, but this concept is not elaborated upon. Further explanation would be helpful.

Reply to Comment 9

We thank you the Reviewer for this comment. In our context, the term “multimodal” was intended to describe the integration and simultaneous resolution of heterogeneous analysis tasks within a unified representation framework, rather than the fusion of data from different acquisition modalities. These tasks include multiscale clustering, clustering of differently shaped structures, and node-level classification.

Traditional clustering pipelines typically address such tasks independently. In contrast, MIRO learns a versatile latent representation that generalizes across these tasks without requiring structural changes, offering a coherent and efficient solution.

In light of the Reviewer’s feedback, we recognize that the term “multimodal” may lead to confusion. We have therefore replaced it with “multifunctional” throughout the manuscript. We have also revised the relevant section to clarify this concept explicitly and to emphasize how MIRO accommodates this functional diversity. The updated text reads:

Additionally, the recurrent structure of MIRO facilitates a multifunctional representation framework, enabling the simultaneous handling of heterogeneous analysis tasks, such as multiscale clustering, clustering of differently shaped structures, and node-level classification. Unlike traditional pipelines that treat these tasks separately, MIRO learns a flexible representation that supports diverse objectives in a unified and scalable manner. This multifunctional capability significantly expands the range of biologically relevant insights that can be extracted from a single experiment.

Comment 10

Minor Comments:

6. Line 109: The phrase “relational information” appears to refer to physical proximity among localizations. The sentence should explicitly state this for clarity.

Reply to Comment 10

In the current implementation, the term “relational information” indeed refers primarily to spatial proximity, represented through Euclidean distances and directional displacement vectors between node pairs.

However, we deliberately use the broader term “relational information” to emphasize the flexibility and extensibility of the MIRO framework. Specifically, MIRO is designed to accommodate a wide variety of edge attributes beyond spatial cues, which makes it applicable to a broader class of problems.

We have clarified this point in the Methods section of the revised manuscript while preserving the general terminology to reflect the method’s broader applicability. The updated sentence now reads:

This selection of node and edge features allows MIRO to inherently analyze graphs of varying sizes and spatial extents without requiring additional processing complexity. Moreover, the architecture is agnostic to the specific type or number of descriptors used: while distance and directional cues serve as the primary relational information in our current

implementation, the framework can readily incorporate additional edge attributes — such as temporal proximity, semantic labels, topological relations, or domain-specific measurements — depending on the needs of the application.

Comment 11

7. Line 157: How is the direction vector encoded in the edge features? Does this imply that the input graph is directed? If so, how is the direction of edges determined, and what is its significance?

Reply to Comment 11

The input graph is undirected, meaning that information flows symmetrically between connected nodes. However, to include directional information in the edge features, we encode the displacement vector from one node to another as an edge attribute. Specifically, for each edge (i, j) , the edge feature includes the vector $\mathbf{d}_{ij} = \mathbf{x}_j - \mathbf{x}_i$, where \mathbf{x}_i and \mathbf{x}_j are the coordinates of nodes i and j , respectively. Since the graph is undirected, each edge is treated symmetrically, and we compute both \mathbf{d}_{ij} and $\mathbf{d}_{ji} = -\mathbf{d}_{ij}$ depending on the message-passing direction. This allows the model to distinguish the relative positions of neighboring nodes while maintaining the undirected structure of the graph. We have clarified this aspect in the revised manuscript to avoid ambiguity. It now reads:

Edge features $\mathbf{e}_{ij} \in E$ encode relational attributes between nodes i and j , such as the Euclidean distance and positional displacement describing their relative arrangement. In the current implementation, each edge feature includes the vector $\mathbf{d}_{ij} = \mathbf{x}_j - \mathbf{x}_i$, where \mathbf{x}_i and \mathbf{x}_j are the coordinates of nodes i and j , respectively. Although the input graph is undirected — meaning that information flows symmetrically between connected nodes — directional information is preserved by assigning displacement vectors \mathbf{d}_{ij} and $\mathbf{d}_{ji} = -\mathbf{d}_{ij}$ to each message-passing direction. This formulation enables MIRO to capture relative orientations of neighboring nodes while maintaining the symmetry of the graph structure.

Comment 12

8. Line 165: It should be explicitly stated that the hidden graph is initialized with node and edge features set to zero.

Reply to Comment 12

As correctly pointed out, the hidden graph is initialized with node and edge features set to zero. We have now further clarified it earlier in the main text at line 177 (note that it was already already stated in the Methods section at line 751, corresponding to line 803 in the revised version):

At each recurrent step, the graph \mathcal{G} is concatenated with a “hidden” graph \mathcal{G}_h^k having the same structure and with node and edge features initialized to zeros.

Comment 13

9. Throughout the manuscript, the authors use the term “linear transformation” to describe data transformations. However, since these transformations are followed by a ReLU activation, they should be referred to as “nonlinear transformations.”

Reply to Comment 13

While it is common practice in the machine learning community to refer to the underlying affine operation (i.e., weight matrix multiplication and bias addition) as a “linear transformation”, even when followed by a nonlinearity such as ReLU, we agree with the Reviewer that this can be misleading in a strict mathematical sense. To enhance clarity and precision, we have revised the manuscript to replace “linear transformation” with “dense layer” and added “followed by ReLU activation” wherever appropriate.

Comment 14

10. Line 178: The manuscript should clarify that the loss is calculated at each recurrent step, rather than at each iteration, training step, or epoch, to avoid potential misunderstandings.

Reply to Comment 14

As in standard training protocols, the loss is computed and used to update the gradients at each iteration (i.e., for each batch) within an epoch. In our case, the model produces intermediate outputs at each recurrent step, and we compute the loss at each of these steps. The final loss for a given iteration is the average of the losses across all recurrent steps and across all batch elements. We have clarified this description in the revised manuscript to prevent potential misunderstandings. It now reads:

To ensure a meaningful hidden representation and prevent vanishing gradients, at each iteration within an epoch, the loss is further averaged across all recurrent steps [30], as schematically shown in Figure 1c.

Comment 15

11. Line 256: The authors suggest using visual inspection of MIRO’s output to guide DBSCAN parameter selection. This process should be described in more detail.

Reply to Comment 15

For consistency with the benchmark, we have selected DBSCAN parameters after MIRO’s transformation with an optimization strategy similar to the one used in the benchmark, which has been automatized with Optuna Python library. This ensures a consistent and unbiased evaluation of the method’s performance. We have updated the text accordingly and removed references to visual inspection. It now reads:

For the benchmark datasets, we used the DBSCAN parameters provided in Ref. [23]. For

MIRO-preprocessed data and other datasets, clustering parameters were optimized using an automated procedure based on the Optuna Python library [35], guided by metric-based performance scores. These parameters were consistently applied across all experiments within the same scenario and are summarized in Supplementary Table 1 and Supplementary Table 2. Please refer to “DBSCAN parameter selection” for a discussion of the parameter choice criteria.

Comment 16

12. Line 356: The manuscript states that cluster shape is encoded as node features. However, since shape is a property of the cluster as a whole rather than individual nodes, further clarification is needed on how this information is incorporated at the node level.

Reply to Comment 16

We thank the Reviewer for pointing out this potential ambiguity. While shape is indeed a property of the cluster as a whole, MIRO learns to infer this information at the node level. During training on multishape datasets, the network is supervised not only to predict displacement vectors for spatial clustering but also to assign a shape class label to each node. These labels reflect the local spatial context of each node and, when aggregated across a cluster, allow for reliable inference of the cluster’s overall shape.

To obtain a cluster-level shape label, we take the mode of the predicted node-level class labels within each cluster. This approach enables the model to perform clustering and shape classification simultaneously within the same framework.

To clarify this in the manuscript, we have revised the original sentence as follows:

However, while transforming various structures into compact forms, MIRO can generate additional output features at the node level that can be used, e.g., for simultaneous cluster shape classification.

Additionally, we have expanded the explanation when presenting the results:

MIRO effectively learns to capture these features at the node level. While clustering ensures accurate separation of structures, taking the mode of node-level class predictions within each cluster allows for reliable identification of the corresponding structural type in heterogeneous datasets.

Comment 17

Overall, this is a well-written manuscript with a novel approach to representation clustering. Addressing the points above would further strengthen the clarity and impact of the work.

Reply to Comment 17

We thank the Reviewer for their positive assessment and constructive feedback. We have addressed all the points raised and believe the revisions have improved the clarity and overall quality of the manuscript.

Response to Reviewer 2

Comment 1

In this paper, the authors present MIRO (Multimodal Integration through Relational Optimization), a novel few-shot (or one-shot) geometric deep learning framework leveraging recurrent graph neural networks (rGNNs) to transform point cloud data for more efficient clustering. By learning to “squeeze” localized points around a common center, MIRO preserves the overall structural relationships within a dataset while enhancing the separation between clustered and non-clustered points, thereby simplifying parameter selection for conventional clustering methods like DBSCAN and supporting structures of varying scales simultaneously. This approach proves effective across a range of single-molecule localization microscopy (SMLM) datasets.

However, the authors need to clarify and support the methodological details and underlying rationales—particularly regarding how MIRO maintains robust performance across diverse scales and complex noise patterns—with more concrete evidence.

Reply to Comment 1

We thank the Reviewer for their thoughtful summary and valuable suggestions. We agree that clarifying the methodological rationale and providing additional supporting evidence strengthens the manuscript. In response, we have revised the text to better explain how MIRO maintains robust performance across varying spatial scales and noise conditions.

Comment 2

1. The authors present MIRO primarily as a booster for existing clustering algorithms, yet their evaluation compares only scenarios with and without MIRO in conjunction with DBSCAN. To strengthen the manuscript, it would be fair to benchmark MIRO’s performance against other clustering “boosters” or enhanced variations of DBSCAN, such as HDBSCAN or DBSCAN++, ensuring fair comparison.

Reply to Comment 2

We thank the Reviewer for this suggestion. Our primary aim was to show how MIRO can improve clustering outcomes by enhancing data representations prior to clustering. We chose DBSCAN as the main baseline and downstream algorithm because previous comprehensive benchmarks have demonstrated its superior performance on SMLM data compared to alternative clustering methods.

Regarding DBSCAN++, it is primarily intended to improve computational efficiency rather than clustering accuracy. While it can accelerate execution, it does not address the main challenge MIRO is designed for—namely, improving the robustness and consistency of clustering across heterogeneous morphologies and densities.

As for HDBSCAN, although it has gained popularity in broader data science contexts, it has seen limited adoption in SMLM analysis and was not included in the benchmark study referenced above. Its comparatively lower performance in this setting may explain this omission. Specifically, a recent study (Ref. [27] in the revised manuscript) quantitatively compared DBSCAN, HDBSCAN, and OPTICS

across a variety of clustering scenarios, and found that DBSCAN consistently outperforms the other two across different cluster types.

These findings are further supported by the SEMORE study (Ref. [42]), which reports clustering results using both DBSCAN and HDBSCAN across several simulated datasets. In cases involving structured or anisotropic morphologies, DBSCAN consistently achieves better performance.

Taken together, these results reinforce our decision to use DBSCAN as a strong and widely accepted baseline for evaluating the effect of MIRO.

We have clarified this rationale in the revised Introduction:

More recently, DBSCAN has been shown to achieve significantly higher scores than HDBSCAN [25] and OPTICS [26] across different cluster types [27].

And in the section “MIRO enhances DBSCAN performance”:

To assess the performance gains introduced by MIRO, we compared the results of DBSCAN both with and without MIRO preprocessing. We selected DBSCAN for this comparison due to its top performance in benchmark studies [23,27] and its widespread use in the literature [28].

Comment 3

Going through the literature, some methods tackle similar problems as MIRO (e.g. <https://www.nature.com/articles/s41467-024-46106-0>).

Reply to Comment 3

The article mentioned by the Reviewer describes SEMORE, a method we explicitly cite and discuss in the previous version of our manuscript (Ref. [40], line 668). While both SEMORE and MIRO aim to address challenges in clustering SMLM data, they differ significantly in their approach and role within the analysis pipeline.

SEMORE consists of two modules: a clustering module that relies on conventional algorithms such as DBSCAN to separate clustered localizations from noise, and a morphological fingerprinting module that computes descriptive features. In contrast, MIRO functions as a pre-clustering representation learning framework, transforming spatial graphs to enhance the robustness and consistency of downstream clustering, particularly across varying densities, shapes, and noise levels.

These approaches could therefore be complementary, with MIRO enhancing the quality of the input to clustering algorithms and SEMORE refining and interpreting their outputs. We have clarified this distinction in the revised Discussion section:

SEMORE [42] introduces a different strategy by applying machine learning for morphological fingerprinting of clusters obtained from density-based clustering. In contrast, MIRO focuses on learning robust spatial representations, enabling both improved clustering and simultaneous structural classification. As such, MIRO could serve as a valuable preprocessing

step that complements methods like SEMORE, potentially improving the quality of the point clouds that SEMORE subsequently analyzes.

Comment 4

2. Although DBSCAN requires two hyperparameters, the MIRO algorithm itself has at least two (and in certain scenarios, three) hyperparameters for its loss function, ground-truth displacement parameter k^* , and the cross-entropy weighting factor—resulting in a total of three to five parameters to tune. The authors should provide additional details on MIRO’s training process, including assessing its robustness to different hyperparameter values, typical ranges, training time, and practical considerations like memory usage and runtime.

Reply to Comment 4

We have now included a series of ablation studies that justify the choice of MIRO’s hyperparameters in the Section “Ablation study and hyperparameter selection”. The hyperparameters used in all the cases are reported in Supplementary Table 6. Specifically, we study the effect of the number of recurrent steps (Supplementary Figure 2), the dimensionality of the hidden representation (Supplementary Table 7), and the weights of the loss function (Supplementary Table 8). These studies show that MIRO is robust to variations in these parameters and that the selected values generalize well across datasets.

Regarding k^* , we clarify that this parameter corresponds to the number of recurrent steps and effectively determines the receptive field size of the network. The optimal value of k^* depends on the spatial scale and structural complexity of the dataset. As shown in Supplementary Figure 2, the network performance improves with increasing the number of recurrent steps up to a point and then saturates, with no evidence of oversmoothing. This analysis provides a principled approach for selecting k^* based on the nature of the clustering problem.

In terms of computational considerations, training time and memory usage are consistent across datasets, as all models are trained under similar conditions using a GPU-equipped workstation. We have added this information to the section “Training efficiency and practical considerations” of the revised manuscript to clarify MIRO’s training procedure and practical implementation:

Training time is not substantially affected by the number of recurrent steps, as the model’s capacity remains constant, with a total of 528,386 trainable parameters. On an NVIDIA A100 GPU with 40 GB of available memory, training a MIRO model takes approximately 20 to 60 minutes per dataset, with variation primarily driven by dataset size and graph density. Memory usage remains modest, owing to the model’s shallow architecture and relatively small parameter count.

Comment 5

3. A key advantage proposed by the authors is MIRO’s capacity for single- or few-shot learning, yet there appears to be no direct evidence in the manuscript demonstrating this. Clarification is needed on whether the reported results were obtained under a single-/few-shot strategy or a more conventional training setup. The authors are encouraged to address how the model’s performance scales with reduced training data, discussing any trade-offs in accuracy versus efficiency.

Reply to Comment 5

To provide direct evidence supporting MIRO’s single-/few-shot learning capabilities, we have expanded the explanation in Section “MIRO training and augmentations” and included a new ablation study in Supplementary Table 5. There, we evaluate MIRO’s performance on representative scenarios, comparing models trained with a single cluster ($n = 1$) and with a number of clusters corresponding to 3 FOVs.

Scenario	N	ARI [†]	IoU	JL _c	RMSRE _{N}	RMSE _{x,y}	AMI	ARI _c	ARI
Scenario 5	1	0.560 ± 0.007	0.614 ± 0.005	0.882 ± 0.008	0.54 ± 0.04	3.16 ± 0.05	0.658 ± 0.004	0.62 ± 0.03	0.576 ± 0.011
	300	0.566 ± 0.005	0.612 ± 0.008	0.870 ± 0.012	0.57 ± 0.04	3.16 ± 0.03	0.654 ± 0.005	0.64 ± 0.03	0.560 ± 0.012
Scenario 6	1	0.726 ± 0.009	0.662 ± 0.013	0.94 ± 0.02	0.24 ± 0.02	3.6 ± 0.2	0.788 ± 0.004	0.87 ± 0.02	0.752 ± 0.004
	60	0.724 ± 0.005	0.680 ± 0.010	0.94 ± 0.02	0.21 ± 0.02	3.59 ± 0.11	0.784 ± 0.005	0.864 ± 0.009	0.760 ± 0.007

Supplementary Table 5 Performance metrics comparing single-sample and full-dataset settings for representative scenarios. Each value is reported as mean ± standard deviation over 5 independently trained models.

The results show that MIRO performs robustly even when trained on a single representative structure, with only modest performance gains as more training data are introduced. These findings confirm that MIRO can generalize effectively from minimal supervision, fulfilling the promise of single- and few-shot learning. We have clarified this point in both the main text and supplementary materials.

To clarify the training regime, we have revised the main text:

In our evaluation, MIRO consistently enhances the performance of DBSCAN across all tested scenarios, as shown in Supplementary Table 3 and Supplementary Table 4. While these results are based on few-shot training (three FOVs, comprising 60 to 300 clusters), we also demonstrate that comparable performance can be achieved with single-shot training, as shown in Supplementary Table 5 for representative scenarios.

Comment 6

4. While the choice to benchmark against DBSCAN follows a previous comprehensive study (<https://www.nature.com/articles/s41592-022-01750-6>), the authors only validated MIRO on four of the nine simulation datasets from that work. More information on why these four were selected—and why the remaining five were not included—would help clarify the completeness of the results. Ideally, the authors could expand their evaluation to all nine datasets or provide a strong rationale for their selection criteria.

Reply to Comment 6

In the revised manuscript, we have expanded our evaluation to include all the eighteen simulation scenarios (nine without and nine with blinking) presented in the benchmark study, thereby ensuring

a comprehensive assessment of MIRO’s performance across a wide variety of synthetic clustering challenges. MIRO demonstrates consistent improvement across all scenarios. The updated results are presented in Supplementary Table 3 and Supplementary Table 4.

Comment 7

5. Regarding reproducibility, the authors should consider making an end-to-end pipeline available in their GitHub repository rather than only sharing the final clusters used in the evaluation. Providing the raw data, preprocessing steps, training scripts, and final evaluation metrics would enable other researchers to replicate the study’s findings precisely and further validate MIRO’s effectiveness.

Reply to Comment 7

The GitHub repository already included the MIRO implementation and training scripts. Nonetheless, to further facilitate reproducibility, we have now updated the repository with additional documentation and detailed instructions for running the full pipeline — covering data preprocessing, training, and evaluation. These additions ensure that other researchers can easily reproduce our experiments and validate the effectiveness of MIRO.

Comment 8

The authors have published their code along with a brief tutorial on how to use it. However, the materials provided for replicating the paper’s results only include CSV files containing the final clustering indices and data point locations. There is no accompanying code or pipeline that demonstrates how to load the data, train the model, and replicate the paper’s findings end to end.

Reply to Comment 8

As noted above, our GitHub repository already includes the code for training, inference, and evaluation. Regarding preprocessing, we would like to clarify that our method is designed to operate directly on the localization lists (i.e., x, y coordinates), which are the standard output of localization microscopy software. Therefore, no additional preprocessing is required prior to applying MIRO.

To further assist users, we have expanded the documentation and provided detailed instructions to replicate the entire pipeline end to end, from raw localization tables to clustering outputs and evaluation metrics.

Point-by-point response to Reviewers of *Enhanced Spatial Clustering of Single-Molecule Localizations with Graph Neural Networks*

Pineda et al.

Response to Reviewer 1

Comment 1

The authors have thoroughly addressed all of my previous concerns. I have no further comments and appreciate the improvements made to the manuscript.

Reply to Comment 1

We thank the Reviewer for their positive feedback and appreciation of the improvements made to the manuscript.

Response to Reviewer 2

Comment 1

I appreciate the authors’ response, and here are my comments:

1. Since the MIRO algorithm is trained in a supervised manner with ground-truth labels, even if it serves only as a pre-processing step for a clustering algorithm, it effectively makes MIRO a supervised clustering approach. Therefore, the result should be improved to some extent. However, the SMLM task, as I understand it, is intended to investigate potential clusters without prior labels. In this case, please clarify the rationale for making the approach supervised. In the benchmark study the authors refer to, only 1 out of 7 methods is supervised. If the authors wish to demonstrate the strength of MIRO, they might consider comparing it against some advanced supervised clustering algorithms.

Reply to Comment 1

We thank the Reviewer for their appreciation of our response. Regarding the new comment, we acknowledge that MIRO is indeed a supervised method, as explicitly stated in the manuscript. However, we would like to emphasize several key aspects that motivate and justify this design choice.

First, as the Reviewer correctly notes, the benchmark study we reference includes a supervised method (CAML), which is based on neural networks. Interestingly, CAML does not outperform the top unsupervised algorithms in that benchmark. This highlights a crucial point: **supervision alone does not guarantee improved performance**. Instead, the architecture and inductive biases of the model — such as those embedded in MIRO — are critical to achieving meaningful gains.

Second, while clustering in SMLM is typically described as an unsupervised task, most methods incorporate prior knowledge or assumptions about cluster characteristics (e.g., size, density). In that sense, even “unsupervised” methods often implicitly encode domain-specific information. Moreover, determining whether a field of view contains clusters is typically performed using separate statistical tools. Therefore, the distinction between supervised and unsupervised methods in this context is often more practical than conceptual.

Following the Reviewer’s suggestion, to further support our approach, **we have added a comparison with a recent advanced supervised method based on graph neural networks (MAGIK-S; Ref. [22])**. As shown in Supplementary Table 6, MIRO outperforms MAGIK-S across all clustering metrics. Notably, MAGIK-S does not always surpass DBSCAN on its own, which reinforces our earlier point: **it is not sufficient to apply supervision — the effectiveness of the representation learning strategy is what ultimately determines success**.

The comparison between MIRO, MAGIK-S, and DBSCAN on two representative benchmark scenarios provided in Supplementary Table 6 is referenced in the text and details of the MAGIK-S architecture are described in a dedicated subsection of the Methods:

We further demonstrate that MIRO outperforms recent supervised methods not included in the benchmark, such as an implementation of the GNN-based framework proposed in Ref. [22] (MAGIK-S), as shown in Supplementary Table 6. Details of the implementation are provided in “Comparison with a Supervised Graph-Based Clustering Framework”.

Finally, MIRO offers several practical advantages over other supervised approaches. It can be trained in single- or few-shot settings, significantly reducing the need for annotated data. It also lowers computational costs by simplifying downstream clustering. These properties are especially important in real-world SMLM analysis, where annotations are scarce and efficiency is key.

To clarify these points, we have extended the Discussion with the following text:

Although MIRO is trained in a supervised manner, we note that supervision alone does not guarantee improved clustering results — as shown in the benchmark study, where the supervised method CAML [20] does not outperform the best-performing unsupervised alternatives. MIRO’s superior performance stems not simply from supervision, but also from its tailored architecture and inductive biases, which enhance representation quality and generalization even with limited training data.

This point is further reinforced by our comparison with MAGIK-S (see Supplementary Table 6), an implementation of a recent supervised method based on graph neural networks [22]. While both MIRO and MAGIK-S incorporate relational inductive biases through the use of graph-based models, the two architectures are designed to tackle different tasks. MAGIK-S is structured as a two-step pipeline involving node-level and edge-level classification, ultimately relying on community detection to form clusters. This approach requires a more complex optimization process, longer training times, and a greater amount of labeled data. In contrast, MIRO is designed specifically to enhance latent representations through recurrent graph operations, allowing for efficient single-shot or few-shot learning and fast generalization across scenarios.

Moreover, the architectural biases in MIRO are fundamentally different from those in MAGIK-S. MIRO explicitly decouples topological refinement from semantic input by maintaining access to the original graph structure across recurrent steps, which enables it to build robust, scale-adaptive representations. This design leads to improved clustering performance with significantly lower computational and data requirements.

Comment 2

2. The authors have provided an appropriate explanation to my earlier question. One minor suggestion: perhaps the GitHub repository’s Jupyter notebook could be provided as a fully executed version, so that users can see the expected results directly.

Reply to Comment 2

We thank the Reviewer for this helpful suggestion. We have now updated the GitHub repository to include a fully executed version of the Jupyter notebook. The expected results can now be directly visualized.